# Hallucination Begins Where Saliency Drops

**Xiaofeng Zhang**[*1] , **Yuanchao Zhu**[*1] , **Chaochen Gu**[1‡] , **Xiaosong Yuan**[2] , **Qiyan Zhao**[1]
**Jiawei Cao**[1] , **Feilong Tang**[3] , **Sinan Fan**[2] , **Yaomin Shen**[1] , **Chen Shen**[2] , **Hao Tang**[4]

[1]Shanghai Jiaotong University    [2]Alibaba Group    [3]Monash University
[4]Peking University
[*]Equal contribution    [‡]Corresponding author
framebreak@sjtu.edu.cn

## Abstract

Recent studies have investigated attention dynamics in large vision language models (LVLMs), yet existing methods remain limited in reliably distinguishing hallucinated from correct outputs — primarily because they rely solely on forward-pass attention, ignoring gradient-based signals that reveal how token influence propagates through the model. To bridge this gap, we introduce **LVLMs-Saliency**, an *gradient-aware diagnostic tool* that quantifies the grounding strength of each output token by fusing attention weights with their gradients. Through analysis, we identify a decisive pattern: *Hallucinations occur when prior output tokens shows low saliency to the next token prediction*, indicating a failure of contextual memory. Building on this insight, we propose a dual-mechanism inference-time framework: (1) Saliency-Guided Rejection Sampling (SGRS), which dynamically filters candidate tokens during decoding by rejecting those with saliency below a context-adaptive threshold, thereby preventing coherence-breaking tokens from entering the sequence; and (2) Local Coherence Reinforcement (LocoRE), a lightweight plug-and-play module that strengthens attention from the current token to its most recent outputs, actively counteracting the "forgetting" behavior identified by LVLMs-Saliency. Experimental results demonstrate that our method significantly reduces hallucinations across multiple LVLMs, offering a robust and interpretable solution to improve model reliability. The code can be accessed in `https://https://github.com/zhangbaijin/LVLMs-Saliency`.

## 1 Introduction

Large Vision Language Models (LVLMs) have made significant strides in cross-modal tasks. However, hallucinations remain a key challenge, particularly in visual question answering and image captioning. Current mitigation strategies such as incorporating external knowledge, retraining with additional data Li et al. (2023a); Liu et al. (2023); Park et al. (2024); Ma et al. (2025e;a;d;b; 2024; 2025c) or training-free methods Neo & Chen (2024); Li et al. (2025a;b); Zhang et al. (2025a); Wu et al. (2025a); Liu et al. (2024c); Gong et al. (2024); Zhou et al. (2024); Shang et al. (2024); Min et al. (2024); Liu et al. (2024b); Fang et al. (2025); Wu et al. (2025b). Although the above methods have made great progress, their interpretability is insufficient, especially without a clear explanation of the causes of hallucinations in the autoregressive generative model.

Recent studies on attention sinks have provided new perspectives for understanding hallucinations. For example, OPERA Huang et al. (2024), DOPRA Wei & Zhang (2024), PAI Liu et al. (2024d), FastV Chen et al. (2024b), EAH Zhang et al. (2024a), TAME Tang et al. (2025a) and Farsight Tang et al. (2025b) have revealed the relationship between attention sinks and hallucinations. They prove that when a token continues to attract high attention weights in subsequent tokens, this over-reliance may cause hallucinations in the model output. However, the relationship between attention maps and hallucinated tokens remains inadequately explained. This is because attention maps only reflect the

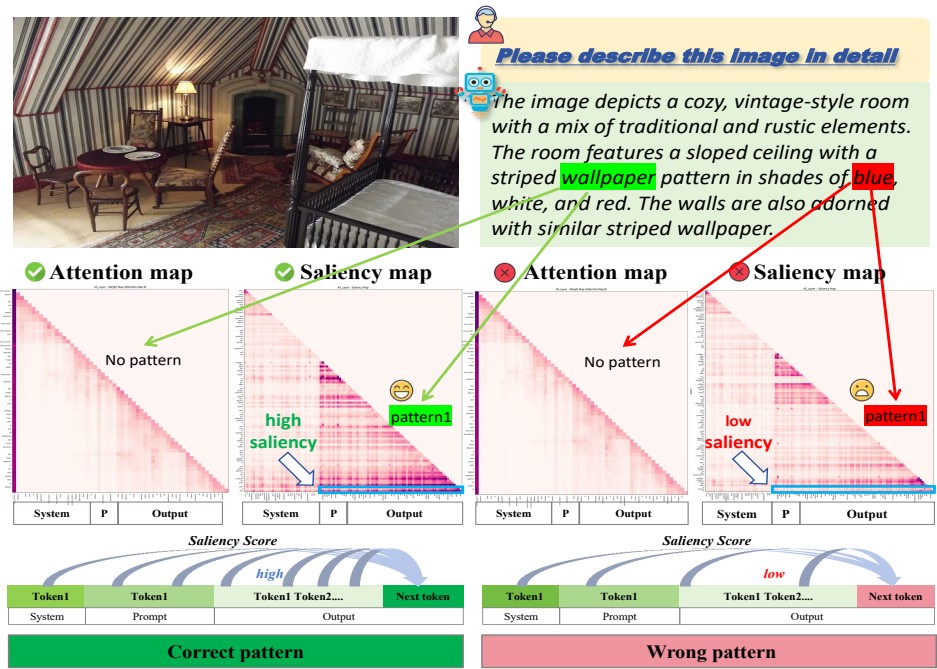

Figure 1: **Attention vs. Saliency Maps for Correct and Hallucinated Tokens (Qwen2-VL-7B).**
Left (correct token **wallpaper**): Attention maps show no distinctive pattern, while our LVLMs-Saliency maps reveal strong, structured grounding to prior outputs. Right (hallucinated token **blue**): Attention maps remain visually similar, but saliency maps collapse, signaling loss of contextual dependency.

model's decision-making in the forward pass, without capturing how changes in input tokens influence the final output. Moreover, existing methods often overlook gradient information, which is essential for understanding the interdependencies among different tokens during the generation process. As illustrated in Figure 1, it is nearly impossible to discern meaningful patterns in attention maps that distinguish correct outputs from hallucinated ones. Therefore, a token-level, interpretable observation tool is essential to uncover the mechanistic origins of hallucinations in large vision-language models, revealing not just when they occur, but why and where in the generation process they emerge.

To address this limitation mentioned above, we draw inspiration from the concept of information flow introduced in "Label Words" Wang et al. (2023), which highlights how information within LLMs tends to converge on specific user-specified tokens. Adapting this insight to the autoregressive generation setting of LVLMs, we propose an **unsupervised metric called LVLMs-Saliency**, defined as the element-wise product of attention weights and their corresponding gradients. This measure quantifies how strongly each previously generated output token influences the prediction of the next token, offering a fine-grained, token-level view of contextual grounding — or its absence — during generation. As shown in Figure 1 and Figure 2, we observe saliency patterns in Qwen2-VL and LLaVA-1.5 that are distinct from conventional attention maps:

> **Pattern:** *Hallucinations occur when prior output tokens shows low saliency to the next token.*

which reveals a breakdown in contextual grounding that attention-only methods fail to capture. When generating the correct token, the model maintains high saliency on previous related tokens, thereby ensuring the coherence of context tokens. However, hallucinations occur when the model "forgets" the past context, resulting in weak dependencies between tokens and low saliency of previous output. By the way, although there is a noticeable difference in the saliency of user prompts for correct versus hallucinated tokens, our analysis of 500 samples indicates that these saliency scores do not significantly affect the model's predictive accuracy. This finding suggests that although prompt saliency plays a role in the model's behavior, it is not the primary cause of hallucinations.

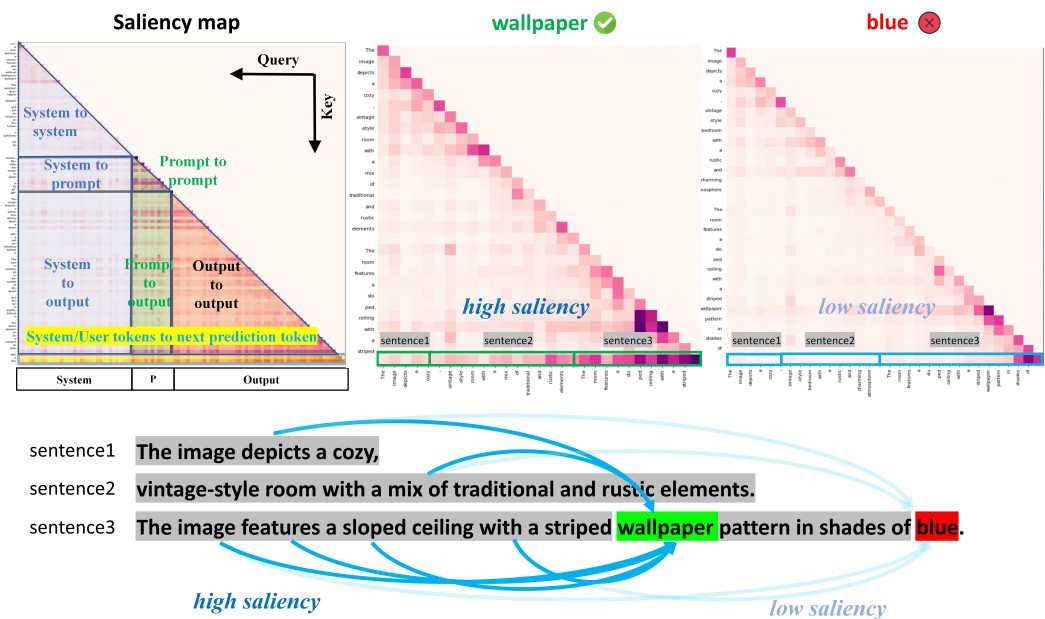

Figure 2: **Output Token Saliency Patterns in Qwen2-VL-7B.** When generating a correct token (e.g., **wallpaper**), the current token assigns high saliency to recent output tokens, typically decaying with distance. In contrast, when generating a hallucinated token (e.g., **blue**), saliency toward all prior outputs collapses — signaling contextual disconnection.

Unlike previous methods of intervening in image attention (Zhang et al., 2024a; Liu et al., 2024d; Jiang et al., 2024; Tang et al., 2025a;b) to alleviate hallucinations, we focus exclusively on the dynamics of **output token saliency** during autoregressive generation. To mitigate hallucinations caused by context loss when the model outputs tokens, we propose a dual-intervention approach in the inference phase that incorporates saliency:

**Saliency-Guided Rejection Sampling (SGRS)**: A proactive filtering mechanism that evaluates the grounding quality of each candidate output token *before* it is committed to the sequence. By computing the token's saliency, SGRS rejects candidates that exhibit weak contextual dependencies (i.e., low saliency), forcing the model to resample until a contextually grounded token is selected. This directly prevents the injection of "coherence-breaking" tokens that trigger cascading hallucinations.

**Local Coherence Reinforcement (LocoRE)**: A reactive stabilization mechanism that activates after a token is accepted. LocoRE strengthens the attention weights from the current query token to the most recent $w_s$ output tokens, using a distance-aware gain factor $\gamma_j^{(P)} = 1 + \beta \cdot \mathbb{I}\left((P - j) \leq w_s\right)$. This ensures that even as the sequence grows, the model maintains strong attentional links to its immediate past, counteracting the "forgetting" behavior observed in Pattern 1.

Together, SGRS and LocoRE form a closed-loop coherence preservation system: SGRS acts as a gatekeeper, blocking low-saliency tokens at the point of entry; LocoRE acts as a stabilizer, reinforcing contextual dependencies after commitment. With extensive experiments, our method demonstrates significant hallucination-mitigating performance across different LVLMs on image hallucination and generation benchmarks, proving its effectiveness. Our contributions are as follows:

- We propose LVLMs-Saliency, an unsupervised, gradient-based metric for quantifying token-level hallucination in autoregressive LVLMs. Through systematic analysis, we establish a direct causal link between low output token saliency and hallucination: when the model fails to maintain attention on recently generated tokens (Pattern 1), contextual memory collapses, leading to semantically inconsistent outputs.
- We introduce Saliency-Guided Rejection Sampling (SGRS), the first inference-time mechanism that dynamically filters candidate tokens based on their saliency with respect to prior output context. By rejecting low-saliency tokens before commitment, SGRS proactively prevents the injection of

coherence-breaking elements into the generation stream — directly mitigating the root cause of context-drift hallucinations.

- We introduce Local Coherence Reinforcement (LocoRE), a lightweight, plug-and-play module that strengthens attention from the current token to its most recent $w_s$ predecessors. Unlike prior methods that rebalance cross-modal attention, LocoRE operates purely within the output stream. SGRS ensures only coherent tokens enter, LocoRE ensures they are not forgotten.

## 2 ANALYSIS AND MOTIVATION

### 2.1 HALLUCINATION TOKEN SALIENCY ANALYSIS

We propose a gradient-based attention analysis framework for quantifying token-level hallucination saliency in autoregressive language models. Given an input sequence $x \in \mathcal{V}^n$, where $\mathcal{V}$ denotes the vocabulary space and $n$ represents the sequence length, we process $x$ through the model $\mathcal{M}$ to obtain:

$$(y, \{\mathbf{A}^{(l,h)}\}_{l=1,h=1}^{L,H}, s) = \mathcal{M}(x), \tag{1}$$

where $\mathbf{A}^{(l,h)} \in [0,1]^{n \times n}$ denotes the attention weight matrix at layer $l \in \{1, \ldots, L\}$ and head $h \in \{1, \ldots, H\}$, $s \in \mathbb{R}^{|\mathcal{V}|}$ represents the logits corresponding to the target hallucination token, $y \in \mathbb{R}^{|\mathcal{V}|}$ is the model's output probability distribution. The cross-entropy loss function $\mathcal{L} : \mathbb{R}^{|\mathcal{V}|} \times \mathbb{R}^{|\mathcal{V}|} \to \mathbb{R}^+$ is defined as:

$$\mathcal{L}(y, s) = -\sum_{t=1}^{T} y_t \log \sigma(s_t), \tag{2}$$

where $\sigma(\cdot)$ denotes the softmax function and $t$ indexes the token position in the sequence. The gradient of the loss with respect to attention matrices is computed as:

$$\nabla \mathbf{A}^{(l,h)} = \frac{\partial \mathcal{L}}{\partial \mathbf{A}^{(l,h)}} \in \mathbb{R}^{n \times n}. \tag{3}$$

The saliency matrix $\mathbf{S}^{(l,h)} \in \mathbb{R}^{n \times n}$ for each attention head is obtained through the Hadamard product followed by triangular masking:

$$\mathbf{S}^{(l,h)} = \mathrm{tril}\left(\left|\mathbf{A}^{(l,h)} \odot \nabla \mathbf{A}^{(l,h)}\right|\right), \tag{4}$$

where $\mathrm{tril}(\cdot) : \mathbb{R}^{n \times n} \to \mathbb{R}^{n \times n}$ preserves the lower triangular portion to maintain causal structure, and $\odot$ denotes element-wise multiplication. The layer-wise normalized saliency $\bar{\mathbf{S}}^{(l)} \in \mathbb{R}^{n \times n}$ is computed by averaging across attention heads and applying $\ell_2$-normalization:

$$\bar{\mathbf{S}}^{(l)} = \frac{\sum_{h=1}^{H} \mathbf{S}^{(l,h)}}{\left\|\sum_{h=1}^{H} \mathbf{S}^{(l,h)}\right\|_2}. \tag{5}$$

As demonstrated in Figures 1, 2, and 5, our quantitative analysis reveals statistically significant differences in saliency patterns between veridical and hallucinated tokens across both Qwen2-VL-7B Yang et al. (2024) and LLaVA1.5-7B Liu et al. (2024a) architectures.

## 3 METHODOLOGY

### 3.1 SALIENCY-GUIDED REJECTION SAMPLING (SGRS)

SGRS dynamically evaluates the grounding quality of each candidate token before commitment; the complete algorithm is formalized in Algorithm 1. At the decoding step corresponding to absolute position $P$, given context $\mathbf{x}_{<P}$ and image $\mathcal{I}$, the model produces logits $s^{(P)} \in \mathbb{R}^{|\mathcal{V}|}$. We sample $K$ candidates $\mathcal{C}^{(P)}$ via top-$K$ sampling. For each $c_i \in \mathcal{C}^{(P)}$, we compute its hallucination saliency $\mathcal{S}(c_i)$ as:

$$\mathcal{S}(c_i) = \frac{1}{|\mathcal{L}_{\text{target}}| \cdot |\mathcal{J}|} \sum_{l \in \mathcal{L}_{\text{target}}} \sum_{j \in \mathcal{J}} \bar{\mathbf{S}}^{(l)}_{P,j}, \tag{6}$$

where $\bar{\mathbf{S}}^{(l)}$ is the layer-wise normalized saliency matrix defined in Section 2.1, $\mathcal{L}_{\text{target}}$ denotes the set of target layers (e.g., middle-to-deep layers), and $\mathcal{J} = \{j \mid \text{Sys}_L + \text{Img}_L \leq j < P\}$ is the set of positions corresponding to previously generated output tokens, with $\text{Sys}_L = 35$ and $\text{Img}_L = 576$ for LLaVA-1.5.

A candidate is accepted only if $\mathcal{S}(c_i) \geq \tau^{(P)}$, where the adaptive threshold is computed over the most recent $W$ output tokens:

$$\tau^{(P)} = \alpha \cdot \frac{1}{|\mathcal{H}|} \sum_{j \in \mathcal{H}} \mathcal{S}(x_j), \quad \mathcal{H} = \{j \in \mathcal{J} \mid (P - 1) - j \leq W\}, \tag{7}$$

with $\alpha \in (0, 1)$ controlling sensitivity and $W$ the history window size. It scales the historical average saliency to control: "How many times the saliency of the current candidate token needs to reach the historical average before it is accepted". If all candidates are rejected, we fall back to selecting the token with the highest saliency score. This mechanism directly operationalizes our finding in Pattern 1: low output-token saliency precedes hallucination. By rejecting such tokens, SGRS enforces a generation path grounded in *textual context* — specifically, the model's own prior outputs.

### 3.1.1 LOCAL COHERENCE REINFORCEMENT (LOCORE)

While SGRS ensures token-level grounding, LocoRe addresses sequence-level context drift by explicitly reinforcing attention dependencies among output tokens, the complete algorithm is formalized in Algorithm 2. Formally, at absolute position $P$ (where $P > \text{Sys}_L + \text{Img}_L$), let $\mathcal{J}_P = \{j \in \mathbb{N} \mid \text{Sys}_L + \text{Img}_L \leq j < P\}$ denote the set of positions corresponding to previously generated output tokens. For the prediction of token at position $P + 1$, we enhance the attention weights from query $P + 1$ to keys in $\mathcal{J}_P$ within a local window of size $w_s$.

Define the distance-weighted gain for each $j \in \mathcal{J}_P$ as:

$$\gamma_j^{(P)} = 1 + \beta \cdot \mathbb{I}\left((P - j) \leq w_s\right), \tag{8}$$

where $\beta \geq 0$ is the reinforcement strength, and $\mathbb{I}(\cdot)$ is the indicator function. Let $\mathbf{A}^{(P+1)} \in \mathbb{R}^{B \times n_h \times (P+1) \times (P+1)}$ denote the attention weight matrix computed during the forward pass for position $P + 1$. We modify the submatrix corresponding to attention from query $P + 1$ to keys in $\mathcal{J}_P$:

$$\mathbf{A}^{(P+1)}[b, h, P + 1, j] \leftarrow \mathbf{A}^{(P+1)}[b, h, P + 1, j] \cdot \gamma_j^{(P)}, \quad \forall b \in [B], \ h \in [n_h], \ j \in \mathcal{J}_P. \tag{9}$$

Equivalently, in vectorized form, let $\boldsymbol{\gamma}^{(P)} \in \mathbb{R}^{|\mathcal{J}_P|}$ be the gain vector with entries $\gamma_j^{(P)}$, and let $\mathbf{A}_{P+1, \mathcal{J}_P}^{(P+1)} \in \mathbb{R}^{B \times n_h \times |\mathcal{J}_P|}$ denote the slice of attention weights from query $P + 1$ to keys in $\mathcal{J}_P$. The update is:

$$\mathbf{A}_{P+1, \mathcal{J}_P}^{(P+1)} \leftarrow \mathbf{A}_{P+1, \mathcal{J}_P}^{(P+1)} \odot \boldsymbol{\gamma}^{(P)}, \tag{10}$$

where $\odot$ denotes element-wise multiplication broadcasted over batch and head dimensions. The modified attention weights are then used in the softmax and weighted sum operations of the self-attention mechanism, ensuring that the model's prediction for token P+1 is more strongly grounded in its recent output history. This operation amplifies the influence of recent context on the prediction of token $P + 1$, directly countering the saliency decay observed in Pattern 1. Crucially, LocoRE operates purely on the attention structure — no gradient computation or model parameter modification is required.

**Synergistic Workflow.** SGRS and LocoRE operate sequentially at each decoding step: SGRS filters and selects the current token $x_P$ based on its saliency to prior outputs; LocoRE then modifies the attention weights used in the *next* forward pass (for position $P + 1$) to reinforce dependencies on recent tokens. This closed-loop design ensures that each accepted token is both well-grounded (SGRS) and unlikely to be forgotten (LocoRE).

**Algorithm 1** SGRS

**Require:** $\mathcal{M}, \mathbf{x}, K, R, \alpha, W, \mathcal{L}, S{=}35, I{=}576, H$
**Ensure:** $x_P$: accepted token at position $P$
1: logits $\leftarrow \mathcal{M}(\mathbf{x}_{\text{input}}, \mathbf{KV})[:, -1, :]$
2: $\mathcal{C} \leftarrow \text{TopK}(\text{softmax}(\text{logits}), K)$, accepted $\leftarrow$ False
3: **for** $r = 1$ to $R$ **do**
4: $\quad c \sim \text{Sample}(\mathcal{C})$
5: $\quad \mathcal{S}(c) \leftarrow \text{SALIENCY}(\mathcal{M}, c, \mathcal{L}_{\text{target}}, P, S, I)$ $\triangleright$ Eq. (1)
6: $\quad \mathcal{J}_P \leftarrow \{j \mid S + I \le j < P\}$ $\triangleright$ Output token positions
7: $\quad \mathcal{H}_P \leftarrow \{j \in \mathcal{J}_P \mid (P-1) - j \le W\}$ $\triangleright$ Recent $W$ outputs
8: $\quad \tau \leftarrow \alpha \cdot \frac{1}{|\mathcal{H}_P|} \sum_{j \in \mathcal{H}_P} H[j]$ $\triangleright$ Eq. (2)
9: $\quad$ **if** $\mathcal{S}(c) \ge \tau$ **then**
10: $\quad\quad x_P \leftarrow c$, $H$.append($\mathcal{S}(c)$), accepted $\leftarrow$ True, **break**
11: $\quad$ **else**
12: $\quad\quad \mathcal{C} \leftarrow \mathcal{C} \setminus \{c\}$
13: $\quad$ **end if**
14: **end for**
15: **if** not accepted **then**
16: $\quad x_P \leftarrow \arg\max_{c \in \text{original } \mathcal{C}} \mathcal{S}(c)$ $\triangleright$ Fallback: best saliency
17: **end if**
18: **return** $x_P$

**Algorithm 2** LOCORE

**Require:**
1: $\mathbf{A}^{(P+1)} \in \mathbb{R}^{B \times n_h \times (P+1) \times (P+1)}$: attention weights for step $P+1$
2: $S = 35, I = 576$: system and image token lengths
3: $w_s$: local window size, $\beta \ge 0$: gain strength
**Ensure:** $\mathbf{A}^{(P+1)}$: modified attention weights for step $P+1$
4: $P \leftarrow$ current position $\triangleright$ Last generated token position
5: $t \leftarrow P - (S + I)$
6: **if** $t \le 0$ **then return** $\mathbf{A}^{(P+1)}$
7: **end if** $\triangleright$ No output yet
8: $\mathcal{J}_P \leftarrow \{j \mid S + I \le j < P\}$ $\triangleright$ Historical output positions
9: **if** $\mathcal{J}_P = \varnothing$ **then return** $\mathbf{A}^{(P+1)}$
10: **end if**
11: **for all** $j \in \mathcal{J}_P$ **do**
12: $\quad d_j \leftarrow P - j$ $\triangleright$ Distance to current position
13: $\quad \gamma_j \leftarrow 1 + \beta \cdot \mathbb{I}(d_j \le w_s)$ $\triangleright$ Eq. (3)
14: $\quad$ **for all** $b \in [B], h \in [n_h]$ **do**
15: $\quad\quad \mathbf{A}^{(P+1)}[b, h, P+1, j] \leftarrow \mathbf{A}^{(P+1)}[b, h, P+1, j] \cdot \gamma_j$ $\triangleright$ Eq. (4)
16: $\quad$ **end for**
17: **end for**
18: **return** $\mathbf{A}^{(P+1)}$

## 4 EXPERIMENTS

### 4.1 EXPERIMENTAL SETUPS

**Baselines.** To demonstrate the broad applicability of our method in LVLM architecture, we applied and evaluated the latest models, including LLaVA-v1.5-7/13B Liu et al. (2024a), Qwen2-VL-7B Wang et al. (2024) and Intern-VL-7/13B Chen et al. (2024d). This study used the following data sets as evaluation sets, representing the expertise in reducing hallucination and general fields.

**Evaluation Benchmarks.** We conduct evaluations on image benchmarks. For image benchmarks, we assess three categories: (1) Comprehensive benchmarks (LLaVA$^{\text{W}}$ Liu et al. (2024a), MM-Vet Yu et al. (2023), MME Yin et al. (2023); (2) General VQA benchmarks (VizWiz Gurari et al. (2018), ScienceQA Lu et al. (2022); (3) Hallucination benchmarks (POPELi et al. (2023b), CHAIR Rohrbach et al. (2018)).

### 4.2 EVALUATION RESULTS ON HALLUCINATION BENCHMARKS

**CHAIR and POPE Evaluations.** As shown in Table 1, methods for mitigating hallucinations can be broadly categorized into third groups. The first group, including OPERA Huang et al. (2024), DOPRA Wei & Zhang (2024), DOLAChuang et al. (2023), VCD Leng et al. (2024), HALC Chen et al. (2024c), An et al. (2024), ICD Zhang et al. (2023), RITUAL Woo et al. (2024), AGLA An et al. (2024), SID Huo et al. (2025), Only Wan et al. (2025), focuses on modifying the decoding process to address hallucinations. The second group, represented by SFT methods such as LESS is more Yue et al. (2024), CCA-LLaVA Xing et al. (2024) and Reverse-VLM Wu et al. (2025b), adjusts the logits of the end-of-sequence (EOS) symbol to control its positioning, allowing the model to terminate earlier, thus reducing hallucinations. The third group includes Vissink Kang et al. (2025), EAH Zhang et al. (2024a), TAME Tang et al. (2025a), MemVR Zou et al. (2024) and Farsight Tang et al. (2025b), which aim to enhance the truthfulness of the model's output during inference by adjusting attention heads. Among these methods,reaching SOTA on the POPE dataset, and achieved significant results second only to EAH on descriptive datasets such as CHAIR. Compared with EAH's approach

Table 1: **Compare results of LocoRE with other SOTA methods on POPE, CHAIR and MME datasets**. The best performances within each setting are **bolded**, baseline: LLaVA-1.5-7B.

| Method | Venue | POPE F1↑ | Acc↑ | CHAIR $C_S$↓ | $C_I$↓ | Recall↑ | length | MME Exist.↑ | Count↑ | Pos.↑ | Color↑ | Total↑ |
|---|---|---|---|---|---|---|---|---|---|---|---|---|
| Beam Search | - | 85.4 | 84.0 | 51.0 | 15.2 | 75.2 | 102.2 | 175.67 | 124.67 | 114.00 | 151.00 | 565.34 |
| Dola Chuang et al. (2023) | ICLR 2024 | 80.2 | 83.1 | 57.0 | 15.2 | 78.2 | 97.5 | 180.10 | 127.40 | 119.30 | 154.60 | 594.10 |
| VCD Leng et al. (2024) | CVPR 2024 | 85.3 | 85.0 | 51.0 | 14.9 | 77.2 | 101.9 | 184.66 | 137.33 | 128.67 | 153.00 | 603.66 |
| OPERA Huang et al. (2024) | CVPR 2024 | 84.2 | 85.2 | 47.0 | 14.6 | 78.5 | 95.3 | 180.67 | 133.33 | 111.67 | 123.33 | 549.00 |
| DOPRA Wei & Zhang (2024) | MM 2024 | 84.6 | 84.3 | 46.3 | 13.8 | 78.2 | 96.1 | 185.67 | 138.33 | 120.67 | 133.00 | 577.67 |
| HALC Chen et al. (2024c) | ICML 2024 | 83.9 | 84.0 | 50.2 | 12.4 | 78.4 | 97.2 | 190.00 | 143.30 | 128.30 | 160.00 | 621.60 |
| CCA-LLaVA Xing et al. (2024) | NeurIPS 2024 | 86.4 | 86.5 | 43.0 | 11.5 | 80.4 | 96.6 | 190.00 | 148.33 | 128.33 | 153.00 | 641.66 |
| RITUAL Woo et al. (2024) | Arxiv 2024 | 85.2 | 84.3 | 45.2 | 13.2 | 78.3 | 99.2 | 187.50 | 139.58 | 125.00 | 164.17 | 616.25 |
| EAH Zhang et al. (2024a) | EMNLP 2025 | 85.7 | 86.0 | 36.4 | 9.9 | 74.9 | 97.7 | 190.00 | 108.33 | **145.00** | 160.66 | 603.99 |
| SID Huo et al. (2025) | ICLR 2025 | 85.6 | 85.8 | 44.2 | 12.2 | 73.0 | 99.4 | 183.90 | 132.20 | 127.80 | 155.90 | 599.80 |
| TAME Tang et al. (2025a) | ICLR 2025 | 85.4 | 85.7 | 41.3 | 12.2 | 74.4 | 98.8 | 193.00 | 137.33 | 139.00 | 164.67 | 634.00 |
| Vissink Kang et al. (2025) | ICLR 2025 | 86.0 | 86.5 | 52.4 | 14.5 | 79.1 | 103.0 | 190.00 | 148.33 | 138.33 | 155.00 | 631.33 |
| CausalLLM Zhou et al. (2025) | ICLR 2025 | 86.0 | 86.5 | - | - | - | - | 195.00 | 156.00 | 135.00 | 170.00 | 656.00 |
| AGLA An et al. (2024) | CVPR 2025 | 84.6 | 85.5 | 43.0 | 14.1 | 78.9 | 98.8 | **195.00** | 153.89 | 129.44 | 161.67 | 640.00 |
| FarsightTang et al. (2025b) | CVPR 2025 | - | - | 41.6 | 13.2 | 75.5 | 100.6 | - | - | - | - | - |
| MemVR Zou et al. (2024) | ICML 2025 | **87.1** | 87.4 | 46.6 | 13.0 | 80.8 | 99.6 | 190.00 | 155.00 | 133.33 | 170.60 | 648.30 |
| ONLY Wan et al. (2025) | ICCV 2025 | 85.5 | 85.1 | 49.8 | 14.3 | 75.9 | 99.7 | 191.67 | 145.55 | 136.66 | 161.66 | 635.55 |
| Reverse-VLM Wu et al. (2025b) | NeurIPS 2025 | - | - | 35.3 | 9.3 | 75.2 | 70.4 | - | - | - | - | - |
| **LocoRE** | - | 86.9 | 87.3 | 38.4 | 11.2 | 75.4 | 98.2 | 190.00 | 158.33 | 133.33 | **175.00** | 656.66 |
| **SGRS + LocoRE** | - | 87.0 | **87.5** | 35.6 | 8.2 | 75.4 | 98.2 | **195.00** | 158.33 | 140.00 | **175.00** | **668.33** |

Table 2: **Comparison of different LVLMs and LocoRE across all image benchmarks**. Notably, in the Hallucination Benchmark, lower scores on CHAIR$_I$ and CHAIR$_S$ indicate better performance, while higher scores are preferable for other metrics.

| Method | Comprehensive Benchmark LLaVA$^W$ | MM-Vet↑ | General VQA VizWiz↑ | SQA↑ | Hallucination Benchmark CHAIR$_S$ ↓ | CHAIR$_I$ ↓ | POPE-R↑ | POPE-F1↑ | POPE-A↑ |
|---|---|---|---|---|---|---|---|---|---|
| LLaVA-1.5-7B | 72.5 | 30.5 | 48.5 | 65.5 | 48.0 | 13.9 | 87.0 | 85.4 | 84.0 |
| +ICD | 69.7 | 30.4 | 46.9 | 62.8 | 47.7 | 13.6 | 87.9 | 84.9 | 84.0 |
| +VCD | 70.9 | 29.5 | 43.4 | 63.3 | 46.8 | 13.2 | 87.0 | 85.3 | 85.0 |
| +OPERA | 72.0 | 31.4 | 50.0 | 64.9 | 45.2 | 12.7 | 88.8 | 84.2 | 85.2 |
| +SID | 73.4 | 31.2 | 50.9 | 67.8 | 44.2 | 14.0 | 89.4 | 85.6 | 85.8 |
| +TAME | 73.9 | 30.5 | 51.6 | 66.0 | 41.3 | 12.2 | 88.9 | 85.4 | 85.7 |
| +Vissink | 74.1 | 33.5 | 53.8 | 67.0 | 52.4 | 14.5 | 87.7 | 84.9 | 85.8 |
| +FarSight | 74.7 | 32.5 | 50.8 | 67.4 | 41.6 | 13.2 | 90.5 | 85.5 | 85.8 |
| **+LocoRE** | **74.8** (+2.3) | **33.8** (+3.3) | **54.8** (+6.3) | **67.5** (+2.0) | 38.4 (+9.6) | **10.2** (+3.7) | 89.5 (+2.5) | 86.9 (+1.5) | 87.3 (+3.3) |
| **+SGRS+LocoRE** | **76.7** (+4.2) | **36.0** (+5.5) | **54.9** (+6.4) | **67.8** (+2.3) | 35.6 (+12.4) | **8.2** (+5.7) | 89.8 (+2.8) | 87.0 (+1.6) | 87.5 (+3.5) |
| LLaVA-1.5-13B | 72.5 | 36.1 | 60.5 | 71.6 | 47.2 | 13.6 | 82.5 | 86.6 | 87.2 |
| **+ LocoRE** | 74.0 (+1.5) | 38.4 (+2.3) | 62.1 (+1.6) | 72.5 (+0.9) | 43.8 (+3.4) | 12.8 (+0.8) | 87.8 (+5.3) | 87.7 (+1.1) | 87.4 (+0.2) |
| **SGRS + LocoRE** | 76.8 (+4.3) | 42.0 (+5.9) | 64.0 (+3.5) | 75.5 (+3.4) | 39.8 (+7.4) | 8.8 (+4.8) | 88.0 (+5.5) | 88.1 (+1.5) | 87.6 (+0.4) |
| Intern-VL-7B | 51.6 | 31.2 | 51.7 | 66.2 | 46.6 | 12.4 | 80.0 | 85.3 | 86.2 |
| **+ LocoRE** | 52.8 (+1.2) | 33.7 (+2.5) | 54.5 (+2.8) | 66.4 (+0.2) | 40.2 (+6.4) | 10.5 (+1.9) | 85.8 (+5.8) | 87.2 (+1.9) | 87.3 (+1.1) |
| **SGRS + LocoRE** | 55.5 (+3.9) | 35.0 (+5.0) | 56.2 (+4.5) | 67.9 (+1.7) | 34.4 (+12.2) | 7.5 (+3.9) | 86.0 (+6.0) | 87.6 (+2.3) | 87.7 (+1.5) |
| Intern-VL-13B | 53.2 | 33.7 | 47.4 | 70.1 | 45.4 | 12.7 | 82.8 | 86.4 | 86.9 |
| **+ LocoRE** | 54.1 (+0.9) | 35.4 (+1.7) | 50.1 (+2.7) | 70.4 (+0.3) | 43.6 (+1.8) | 12.5 (+0.2) | 86.3 (+3.5) | 87.2 (+0.8) | 87.3 (+0.4) |
| **SGRS + LocoRE** | 56.8 (+3.6) | 37.3 (+3.6) | 52.0 (+4.6) | 71.0 (+0.9) | 45.2 (+3.4) | 14.0 (+2.7) | 87.0 (+4.2) | 88.1 (+1.7) | 88.8 (+1.9) |
| Qwen2-VL-7B | 75.6 | 63.2 | 57.3 | 74.1 | 25.0 | 7.3 | 79.1 | 86.6 | 87.6 |
| **+ LocoRE** | 77.8 (+2.2) | 64.8 (+1.6) | 59.4 (+2.1) | 74.2 (+0.1) | 23.5 (+1.5) | 6.8 (+0.5) | 81.3 (+2.2) | 87.5 (+0.9) | 88.2 (+0.6) |
| **SGRS + LocoRE** | 79.7 (+4.1) | 67.7 (+4.5) | 60.3 (+3.0) | 75.3 (+1.2) | 19.3 (+5.7) | 5.1 (+2.2) | 82.6 (+3.5) | 88.0 (+1.4) | 89.0 (+1.4) |
| Qwen2.5-VL-7B | 76.8 | 62.2 | 60.9 | 79.0 | 27.2 | 9.0 | 80.4 | 87.4 | 88.4 |
| **+LocoRE** | 77.9 (+1.1) | 64.8 (+2.6) | 61.6 (+0.7) | 80.8 (+1.8) | 23.0 (+4.2) | 8.5 (+0.5) | 80.9 (+0.5) | 87.8 (+0.4) | 88.7 (+0.3) |
| **SGRS +LocoRE** | 80.0 (+3.2) | 66.2 (+4.0) | 62.7 (+1.8) | 82.1 (+3.1) | 21.0 (+6.2) | 6.5 (+2.5) | 81.5 (+0.5) | 88.3 (+0.9) | 89.5 (+1.1) |
| Qwen2.5-VL-32B | 81.2 | 72.2 | 70.8 | 89.0 | 43.6 | 9.5 | 79.1 | 86.7 | 87.8 |
| **+LocoRE** | **82.7** (+0.5) | **73.1** (+0.9) | **71.2** (+0.4) | **89.3** (+0.3) | 41.8 (+1.8) | **8.5** (+1.0) | 79.5 (+0.4) | 86.9 (+0.2) | 88.0 (+0.2) |

of directly replacing the attention head, LocoRE has a higher recall because it does not change the internal representation of the model, and therefore does not affect the diversity of the model output.

Compared to Vissink Kang et al. (2025) and TAME Tang et al. (2025a), which also allocate attention, LocoRE's CHAIR performance is more prominent. TAME allocates the attention on the system token to other tokens, but still ignores the visual information, while Vissink only intervenes with the visual attention sink and ignores the contextual association of the text output. As a result, both of them perform not that well on long text output datasets such as CHAIR, while this also proves the effectiveness of our approach, which is able to address the shortcomings of both of them, i.e., enhancing the visual information as well as enhancing the contextual dependencies between text outputs.

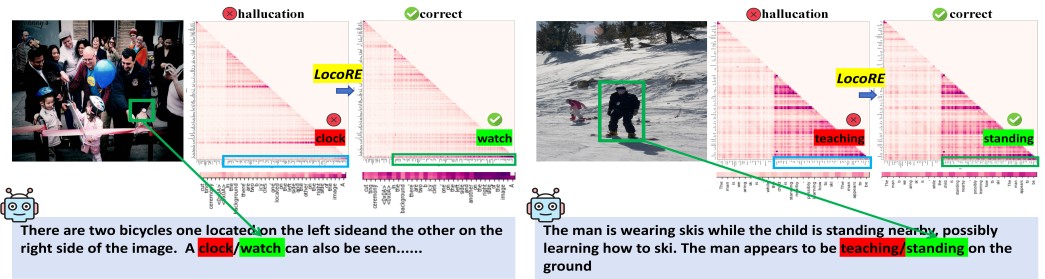

Figure 3: **Effect of LocoRE on output token saliency map (Qwen2-VL-7B).** *Without LocoRE*: When generating an incorrect token(**clock**), saliency scores assigned to prior output tokens are low — indicating weak contextual grounding. *With LocoRE*: The same position now generates a correct token(**watch**), accompanied by significantly higher saliency scores to recent outputs — demonstrating LocoRE's ability to restore contextual coherence and prevent hallucination via attention reinforcement.

### 4.3 EVALUATION RESULTS ON GENERATION BENCHMARK

**MME and Other Benchmarks Evaluations.** As shown in Table 1 and Table 2, we tested on several popular LVLMs' general ability benchmarks. MME comprises ten subtasks to evaluate models' perceptual capabilities and four subtasks for assessing recognitive abilities in the form of yes/no questions. LocoRE can maintain and improve the multimodal capability on LVLMs benchmarks. Our method achieve a much higher score (corresponds to less hallucination) across all categories. This underscores its effectiveness in addressing a broader range of multimodal hallucination challenges beyond objects. Combining SGRS with LocoRE further improves reasoning-intensive tasks, as demonstrated by the cognitive categories of MME. This performance is particularly pronounced on the "**Existence**" and "**Position**" tasks, as SGRS directly suppresses hallucinations while LocoRE focuses solely on contextual coherence.

### 4.4 ABLATION STUDY

**Effect of LocoRE on other LVLMs** As shown in Table 2, the integration of LocoRE as a plug-in into LLaVA-1.5-7B/13B Liu et al. (2024a), Qwen2-VL-7B/13B/32B Wang et al. (2024) and Intern-VL-7/13B Chen et al. (2024d), was effective in improving results in both integrated and generalized VQA tasks. In addition, it achieved a significant improvement in hallucination metrics. These results indicate that LocoRE is effective in reducing hallucinations in both structured and unstructured environments.

**Saliency map Visualization with LocoRE.** As shown in Figure 3, which visualizes the LVLMs-Saliency maps from prior output tokens to the current token, applying LocoRE significantly increases the saliency scores assigned to recently generated context tokens — particularly those within the local coherence window. This demonstrates that LocoRE effectively strengthens the model's dependency on its immediate output history, counteracting the "forgetting" behavior observed in the baseline. The saliency boost under LocoRE confirms our design principle: by explicitly reinforcing attention to recent outputs, the model maintains stronger contextual links during autoregressive generation. This prevents the decay of intra-output saliency that leads to hallucinations, ensuring that each new token remains grounded in its textual predecessors.

### 4.5 ABLATION STUDY ON KEY HYPERPARAMETERS

We evaluate $\alpha$ (SGRS) and $\beta$ (LocoRE) on both CHAIR and POPE benchmarks. As shown in Table 3 and Figure 4, our full method ($\alpha = 0.6, \beta = 0.15$) reduces CHAIR hallucination rate by 28.3% (LLaVA-1.5) and 22.8% (Qwen2-VL) compared to baseline. SGRS alone ($\alpha = 0.6, \beta = 0.0$) contributes most of the improvement, but LocoRE adds further gains (e.g., POPE $F1 - score$ from 85.4% to 86.9% in LLaVA-1.5). Increasing $\alpha$ to 0.9 yields marginal improvement at high latency cost (+33%). We recommend $\alpha = 0.6, \beta = 1.2$ as the optimal balance. While increasing $\alpha$ to 0.9 further reduces hallucination rates (CHAIR$_S$: 35.6% $\rightarrow$ 30.0%; POPE: 87.0% $\rightarrow$ 87.1%), it incurs a 33%

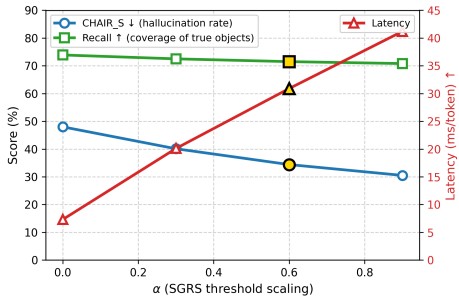

Figure 4: Ablation study of $\alpha$: trade-offs between hallucination rate, recall, and latency.

| $\alpha$ | $\beta$ | LLaVA-1.5 | | | | Qwen2-VL-7B | | | |
|---|---|---|---|---|---|---|---|---|---|
| | | CHAIR | | POPE | | CHAIR | | POPE | |
| | | S↓ | I↓ | F1↑ | Acc↑ | S↓ | I↓ | F1↑ | Acc↑ |
| 0.0 | 0.0 | 48.0 | 13.9 | 85.4 | 84.0 | 25.0 | 7.3 | 86.6 | 87.6 |
| 0.0 | 0.15 | 38.4 | 10.2 | 86.9 | 87.3 | — | — | — | — |
| 0.0 | 0.20 | — | — | — | — | 23.5 | 6.8 | 87.5 | 88.2 |
| 0.6 | 0.0 | 36.5 | 9.0 | 86.9 | 87.4 | 20.5 | 5.6 | 87.9 | 88.9 |
| 0.6 | 0.15 | **35.6** | **8.2** | **87.0** | **87.5** | — | — | — | — |
| 0.6 | 0.20 | — | — | — | — | **19.3** | **5.1** | **88.0** | **89.0** |
| 0.6 | 1.0 | 50.2 | 20.9 | 60.3 | 57.8 | 37.5 | 18.5 | 55.3 | 54.6 |

Table 3: **Ablation study on $\alpha$ (SGRS) and $\beta$ (LocoRE)**. Best in **bold**. $\beta$: 0.15 (LLaVA-1.5), 0.20 (Qwen2-VL).

higher latency cost (30.8 ms/token $\rightarrow$ 41.2 ms/token) and risks degrading generation fluency due to over-rejection. In extreme cases, correct but moderately salient tokens may be rejected, leading to fallback-generated outputs that are less diverse or natural. We thus recommend $\alpha = 0.6$ as the optimal trade-off — it suppresses 28.3%+ of hallucinations while maintaining practical inference speed and output quality.

# 5 RELATED WORK

## 5.1 NEXT TOKEN PREDICTION

After obtaining the next token probability, different decoding strategies are proposed to predict the next token. The decoded token Huang et al. (2024); Chuang et al. (2023); Chen et al. (2024a) is concatenated with the last of the original input text for the next-token generation until the generation ends.

## 5.2 INFORMATION FLOW OF IN LVLMS

Some researchHuang et al. (2024); Wei & Zhang (2024); Zhang et al. (2024b; 2025c;b); Zhao et al. (2025); Zhang et al. (2026); Zhao et al. (2026) uses Grad-CAM and attention maps to visualize the interaction between images and text in complex reasoning tasks. Attention scores highlight relevant areas through forward propagation. The EAH Zhang et al. (2024a) identifies that most hallucinations stem from the attention sink pattern marked by images in the attention matrix. Based on this insight, EAH proposes a method that enhances attention heads without additional training. TAME Tang et al. (2025a) and Farsight Tang et al. (2025b) investigate the causes of hallucinations by analyzing local self-attention patterns of "anchor tokens" and defines the degree of attentional localization as the probability of token propagation.

# 6 CONCLUSION

In this work, we revisit the conventional explanations linking attention sinks to hallucinations and propose a saliency-based framework to complement existing analyses. Our findings reveal that hallucinations frequently correlate with weak saliency in prior output tokens. To this end, we introduce SGRS and LocoRE, a plug-and-play intervention that dynamically boosts visual attention and reinforces local coherence during text generation. Experiments confirm that LocoRE consistently improves output accuracy across various benchmarks without requiring model retraining.

# 7 ACKNOWLEDGMENTS

This work was supported by the National Natural Science Foundation NO. 62273235, National Major Scientific Research Instrument Development Project (62227811).

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

## A  RELATED WORK

### A.0.1  INFERENCE-TIME EFFICIENCY

While our full framework (SGRS + LocoRE) achieves the strongest hallucination suppression, it incurs higher latency due to the backward pass required for saliency computation in SGRS — typically adding 30–40% overhead per token compared to standard greedy decoding.

While the full SGRS+LocoRE framework achieves the strongest hallucination suppression, its reliance on gradient computation introduces non-negligible latency overhead — making it less suitable for real-time applications. In practice, however, **LocoRE alone serves as a highly effective compromise**: as a forward-only module that manipulates attention weights in-place, it incurs <2% latency increase while still significantly mitigating context-drift hallucinations.

As shown in Figure 6, compared to prior plug-and-play methods — such as VCD Leng

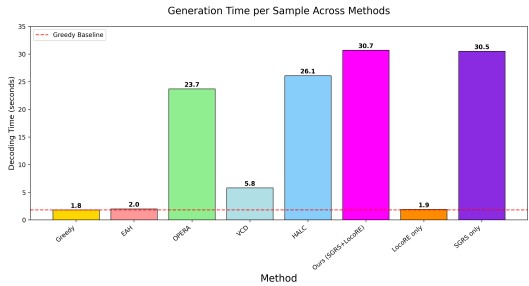

Figure 6: Generation time of a single response.

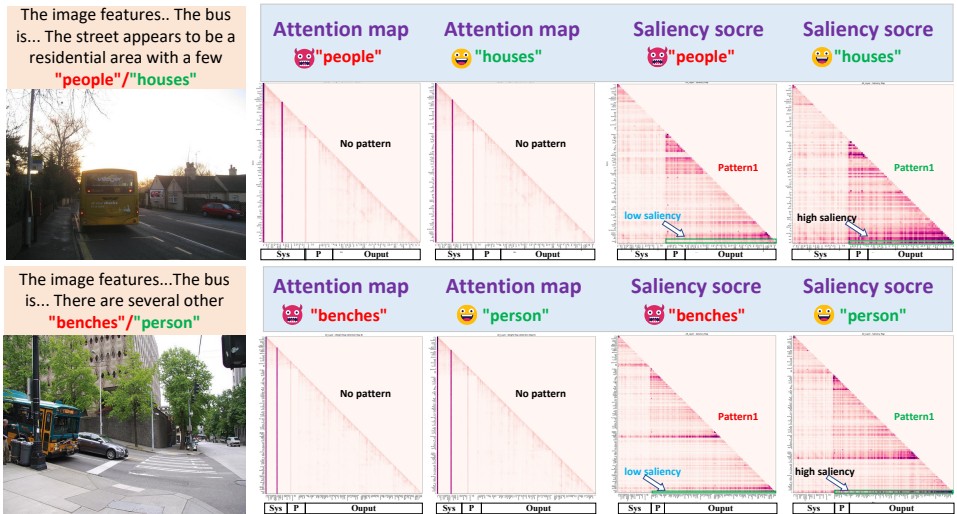

Figure 5: Attention map and saliency map of LLaVA1.5-7B.

et al. (2024), OPERA Huang et al. (2024), Far-
sight Tang et al. (2025b), HALC Chen et al.
(2024c), and EAH Zhang et al. (2024a) — LocoRE requires no auxiliary models, no external detectors, and no multi-pass decoding. By operating entirely within the standard autoregressive loop, it achieves superior speed-efficiency trade-offs.

## A.1 SALIENCY SCORE

To reveal why the MLLM produces a hallucination token, it is necessary to elucidate the information flow. In this section, we use saliency score to analyze the information flow across the different tokens(system, image, prompt, and output). In this section, we examine 4 types of token(system/image/prompt/output). We can use a Taylor expansion to compute the saliency score for each element of the attention matrix:

$$S_l = \left| \sum_h A_{h,l} \odot \frac{\partial \mathcal{L}(x)}{\partial A_{h,l}} \right|, \tag{11}$$

where $A_{h,l}$ denotes the value of the attention matrix for the $A_{h,l}$ attention head in layer $l$, and $x$ denotes the input. $\mathcal{L}(x)$ is the loss function of the task, e.g., the cross-entropy of the quiz task objective. The saliency matrix $S_l$ for layer $l$ is obtained by averaging all heads of attention. More saliency maps and attention maps of LLaVA 1.5/Qwen2-VL are shown in Figure 5 and Figure 13 and Figure 14.

## A.2 INFORMATION FLOW OF IN LLMS

Information flow provides an intuitive method of understanding the internal mechanisms of the black-box models of LVLM. Label words Wang et al. (2023), and ACT Yu et al. (2024) are early works that explore the mechanism of LLMsZhu et al. (2023); Devlin et al. (2018); Touvron et al. (2023) by rving information flow patterns. By calculating saliency scores, it is possible to visualize the information flow.

StreamingLLM Xiao et al. (2023) introduces the concept of attention sink, observing an intriguing phenomenon: Initial tokens, despite their seemingly minor role in content generation, consistently receive high attention scores. This is visualized in the attention map as columns with notably high attention scores, which is counterintuitive. Due to the autoregressive nature of generative models, these initial tokens continue to attract attention from subsequent tokens, amplifying their impact on the generation process. To address this, StreamingLLM leverages attention-sink tokens during the pre-training phase to enhance the model's performance.

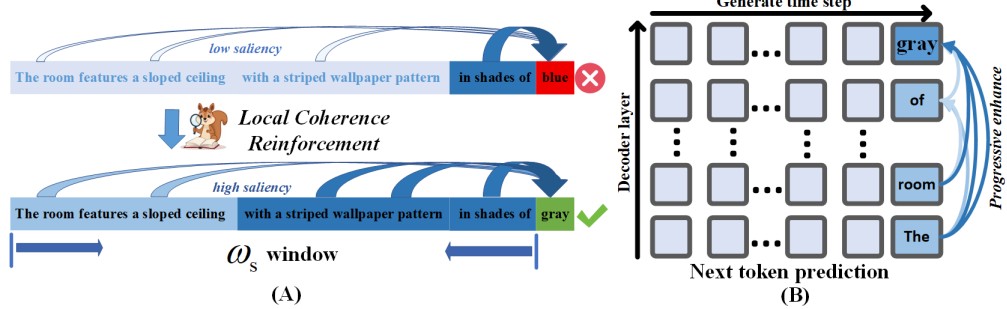

Figure 7: The structure of Local Coherence Reinforcement (LocoRe): attention from the next token to recent outputs is enhanced to preserve contextual coherence.

Massive activationsSun et al. (2024) highlights that, while there are approximately 40,000 activations per hidden state, only four are recognized. In the feature dimension of language models, large activations consistently occur in a very small number of fixed dimensions. LLMs are categorized into three types based on the location of massive activations: (a) occurring only at the onset, (b) occurring at the onset of lexical elements and the first "strong separator" word (e.g. ".", "/n"), or (c) occurring at the onset, separator words (e.g., ".", "/n"), and the first "strong separator" word, as well as some semantically weaker words (e.g., "and", "from", "of").

### A.3 INFORMATION FLOW OF IN LVLMS

LLaVA-CAM Zhang et al. (2024b; 2025c;b); Wei & Zhang (2024) utilizes Grad-CAM and attention maps to visualize the interaction between images and text in complex reasoning tasks. Attention scores highlight relevant areas through forward propagation, while Grad-CAM captures gradient changes through backpropagation, revealing the salience of image features. These complementary approaches provide a comprehensive understanding of the dynamics of information flow by assessing the importance of input and demonstrating their specific impact on model predictions.

The EAH study Zhang et al. (2024a) identifies that most hallucinations stem from the attention sink pattern marked by images in the attention matrix. To address this, EAH proposes a method that enhances attention heads without additional training. By strengthening attention heads with visual depression characteristics in shallow layers, the method improves attention distribution for image tokens, effectively reducing hallucinations across various LVLMs.

TAME Tang et al. (2025a) investigates the causes of hallucinations by analyzing local self-attention patterns of "anchor points" and defines the degree of attentional localization as the probability of token propagation. The analysis reveals that over-propagation of anchor tokens occurs when the eigenvalue distributions of the query and key matrices exhibit a non-zero mean and polarized variance, leading to an over-reliance on anchor tokens while ignoring visual information, resulting in hallucinations.

As illustrated in Figure 8, in summary, EAH Zhang et al. (2024a) differs from existing methods while remaining non-conflicting and even complementary. Existing methods primarily adjust decoding strategies by modifying logits. OPERA Huang et al. (2024) and DOPRA Wei & Zhang (2024) identify that anchor output tokens can lead to hallucinated token generation and try to penalize anchor tokens' logits. TAME Tang et al. (2025a) focuses on the propagation of the anchor token in all layers, dynamically adjusting these anchor tokens.

### A.4 LIMITATIONS

The primary limitation of our Saliency-Guided Rejection Sampling (SGRS) framework lies in its computational demands: computing token-level saliency requires storing intermediate activations and performing backward passes during inference, which consumes significant GPU memory. As a result, we are currently unable to deploy SGRS on very large models such as 72B-parameter LLMs, where gradient computation exceeds the memory capacity of even high-end GPUs (e.g., A100 80GB).

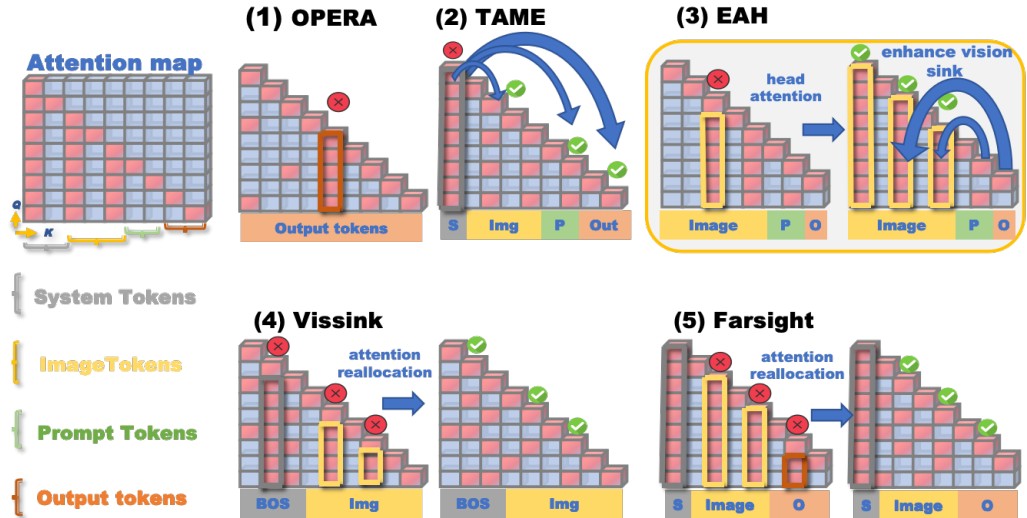

Figure 8: The information flow of different models.

This restricts our evaluation to models up to 7B–13B parameters (e.g., LLaVA-1.5, Qwen2-VL-7B), limiting the generalizability of our full framework to the largest-scale architectures.

However, we emphasize that our **LocoRE module remains fully applicable to any model size**, as it operates purely in the forward pass and requires no gradient computation. For large models where SGRS is infeasible, LocoRE alone provides a lightweight, plug-and-play solution that still significantly mitigates context-drift hallucinations.

## A.5  VIDEO BENCHMARKS EVALUTION

In Zero-Shot Video Question Answering Tasks, LocoRE achieves significant improvements over video MLLM such as Video-LLaVA Lin et al. (2023) and Video-LLaMA2 Cheng et al. (2024) in three key benchmark datasets. As shown in Table 4, on the MSRVTT-QA dataset, our method delivers an average accuracy gain.

Table 4: **Comparison of different Video LVLMs and LocoRE across all video benchmarks**. In the Video-Based Text Generation Benchmark, five scores are assessed: Cr. (Correctness of Information), Cs. (Consistency), De. (Detail Orientation), Ct. (Contextual Understanding) and Te. (Temporal Understanding). Following Maaz et al Maaz et al. (2023), we use the GPT-3.5 Turbo model to assign a relative score to the model outputs, with scores ranging from 0 to 5.

| Method | MSVD-QA | | MSRVIT-QA | | ActivityNet-QA | | Video-Based Text Generation | | | | |
|---|---|---|---|---|---|---|---|---|---|---|---|
| | Accuracy ↑ | Score↑ | Accuracy↑ | Score↑ | Accuracy ↑ | Score↑ | Cr.↓ | Cs.↓ | De.↓ | Ct.↓ | Te.↓ |
| Video-LLaVA | 64.8 | 3.7 | 59.0 | 3.5 | 41.5 | 3.3 | 2.32 | 2.34 | 2.65 | 2.75 | 2.09 |
| + LocoRE (Ours) | 65.9 (+1.1) | 3.8 | 61.3 (+2.3) | 3.5 | 41.9 (+0.4) | 3.5 | 2.36 | 2.42 | 2.88 | 2.87 | 2.12 |
| Video-LLaMA2 | 70.9 | 3.8 | 67.2 | 3.6 | 49.9 | 3.3 | 3.13 | 3.23 | 2.70 | 3.42 | 2.45 |
| + LocoRE (Ours) | 71.8 (+0.9) | 3.9 | 69.9 (+2.7) | 3.7 | 52.2 (+2.3) | 3.6 | 3.36 | 3.41 | 2.91 | 3.55 | 2.66 |

**Differences with other fusion strategies**: We compared the other three fusion strategies:

- Addition: After the attention and gradient are added, the visualized image shows an overall high state, and the pattern cannot be distinguished at all. It is a meaningless pattern. The fusion scores of many tokens are concentrated in the middle range, and the overall appearance is "gray", making it difficult to distinguish the key tokens.
- Maximum value (Max): Take the larger of the attention and gradient. Although it can amplify individual high values, the visualization result still cannot distinguish the effective pattern.
- Concat + MLP: Gradient and attention are spliced and then adaptively fused through the neural network. The score distribution is rich. The visualization is similar to the addition.

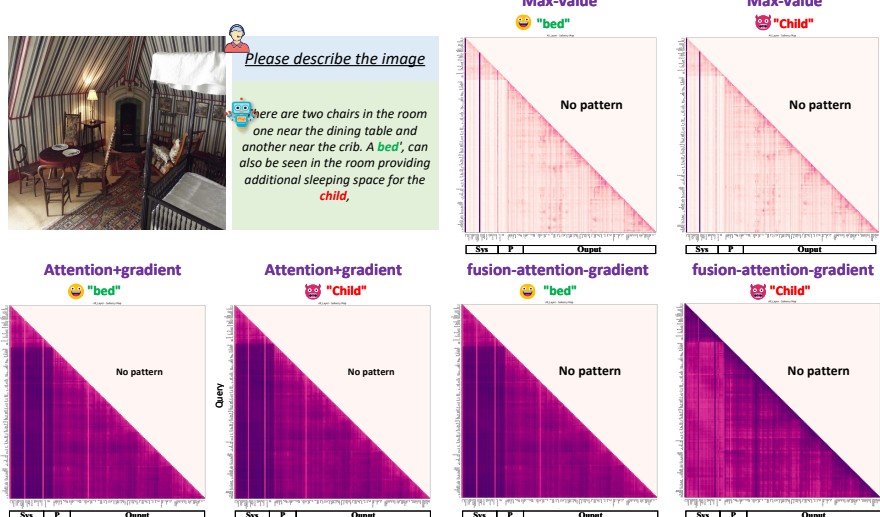

Figure 9: **The saliency map of output token.** We compared the other three fusion strategy including Addition, Maximum value (Max) and concat attention+gradient by mlp, there is no pattern between normal and hallucination token, in contrast, attention*gradient fusion is clearer in distinguishing important from unimportant tokens.

The lower triangle shows a color close to the same, and the effective pattern cannot be distinguished.

**In contrast, attention*gradient fusion is: clearer in distinguishing important from unimportant tokens**.

## USE OF LLM

The authors used generative AI tools (e.g., Grammarly, ChatGPT) solely for grammar checking and language polishing of the manuscript. All technical content, experimental design, data analysis, and conclusions were generated and verified exclusively by the human authors. The use of AI tools does not affect the originality or authorship of this work.

## B    ETHICS STATEMENT

This work focuses on improving the reliability of large vision-language models (LVLMs) by mitigating hallucination through inference-time interventions. Our method, SGRS+LocoRE, operates solely on publicly available models (e.g., LLaVA-1.5, Qwen2-VL) and benchmark datasets (e.g., CHAIR, POPE, MME), without collecting or using any private, sensitive, or human-subject data. The proposed techniques do not introduce new biases beyond those already present in the base models, and they are designed to enhance — not replace — human oversight in critical applications. We acknowledge that while our method reduces hallucination, it does not eliminate all risks of harmful or misleading outputs. Users should exercise caution when deploying LVLMs in high-stakes scenarios such as medical diagnosis, legal advice, or autonomous decision-making.

## C    REPRODUCIBILITY STATEMENT

To ensure full reproducibility, we provide the following resources: (1) **Code**: Complete implementation of SGRS and LocoRE, including saliency computation, rejection sampling, and attention reinforcement modules, will be released publicly on GitHub upon publication. (2) **Hyperparameters**: All key hyperparameters ($\alpha = 0.6$, $\beta = 0.15$ for LLaVA-1.5; $\beta = 0.20$ for Qwen2-VL-7B) and training-free inference protocols are detailed in Section 4.5. (3) **Evaluation**: We use standard,

publicly available benchmarks (CHAIR, POPE, MME) with official evaluation scripts. All results are averaged over 500 samples with fixed random seeds. (4) **Compute**: Experiments are conducted on NVIDIA A100 80GB GPUs; average latency is reported in ms/token (Figure 4). (5) **Models**: We evaluate on open-source LVLMs: LLaVA-1.5 (13B) and Qwen2-VL-7B, using official checkpoints from Hugging Face Model Hub. No proprietary data or models are used in this work.

## D  MORE EXPERIMENTS IN REBUTAL

### D.1  STATISTICAL VALIDATION OF THE SALIENCY-HALLUCINATION RELATIONSHIP

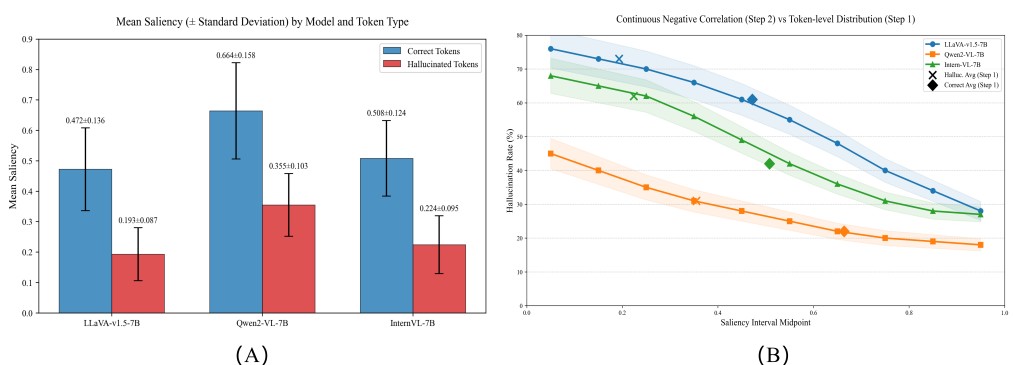

Figure 10: **Statistical analysis of output-token saliency vs. hallucination.** (a) Mean saliency for correct vs. hallucinated tokens across three models. (b) Hallucination probability as a function of saliency bin (per model average).

To rigorously test our core hypothesis: *"The saliency score of hallucination tokens is often relatively low."* — we conduct three complementary quantitative analyses at the token level across three diverse VLMs: LLaVA-v1.5-7B, Qwen2-VL-7B, and InternVL-7B. All experiments are performed on the POPE and CHAIR benchmarks, with hallucination labels assigned via human annotation.

**Token-level saliency distribution: hallucinated vs. correct tokens**. For each generated token $y_t$ in our dataset ($\sim$12,000 tokens total), we compute the saliency score from the immediately preceding output token to the current token. We then group tokens by label (correct or hallucinated) and report mean $\pm$ standard deviation.

As shown in Figure 10(a) and Table 5, a consistent and statistically significant pattern emerges across all models:

Table 5: **Mean saliency scores for correct vs. hallucinated tokens across models.**

| Model | Correct Tokens | Hallucinated Tokens |
|---|---|---|
| LLaVA-v1.5-7B | $0.472 \pm 0.136$ | $0.193 \pm 0.087$ |
| Qwen2-VL-7B | $0.664 \pm 0.158$ | $0.355 \pm 0.103$ |
| InternVL-7B | $0.508 \pm 0.124$ | $0.224 \pm 0.095$ |

These results confirm that the significantly lower saliency scores of hallucinated tokens, compared with correct tokens, is a phenomenon that generalizes across different model architectures.

**(2) Saliency score and negative correlation with hallucination**: As shown in Figure 10(b), we divide the saliency score of the previous output token into 10 equally wide intervals and calculate the conditional probability $P$ of hallucination in each interval. All three models (LLaVA-v1.5-7B, Qwen2-VL-7B, and InternVL-7B) showed a strong negative correlation: the hallucination rate systematically decreased with increasing saliency. A clear, smooth, and monotonic negative correlation is evident: <1> In the lowest saliency $[0.0, 0.1)$, hallucination rates reach **68%–76%**;

Table 6: Hallucination experiments that artificially lower saliency scores

| Decay rate $r$ | CHAIRs ↓ | POPE-F1 ↑ | POPE-A ↑ |
|---|---|---|---|
| 1.0 | 35.6 | 87.0 | 87.5 |
| 0.8 (decay 20%) | 37.9 | 86.5 | 86.8 |
| 0.6 (decay 40%) | 42.1 | 85.4 | 85.6 |
| 0.4 (decay 60%) | 47.8 | 84.8 | 84.0 |
| 0.2 (decay 80%) | 56.0 | 83.0 | 83.8 |

<2> In the highest $[0.9, 1.0]$, rates drop to **18%–28%**. The trend holds across all models, with no non-monotonic jumps or plateaus.

**(3) Saliency Intervention Experiment:** As shown in Table 6, we also conducted an intervention experiment on the LLaVA-v1.5-7B model. For each sample in the POPE and CHAIR datasets, highly significant correct tokens were selected for intervention (these tokens came from the correct tokens with saliency > 0.45 in Step 1). The intervention method was as follows: after generating the target token, its saliency output in the decoder was scaled (multiplied by a factor $r \in 1.0, 0.8, 0.6, 0.4, 0.2$) to simulate the process of its saliency being weakened. The results showed that after the saliency value was artificially reduced, the hallucination rate increased significantly."

**Conclusion. These findings support our claim that hallucinations are not triggered by a single threshold event, but rather emerge gradually as contextual saliency decays. This gradient nature suggests that saliency can serve as a continuous diagnostic signal.**

### D.2 FAILURE CASE: HIGH-SALIENCY HALLUCINATION

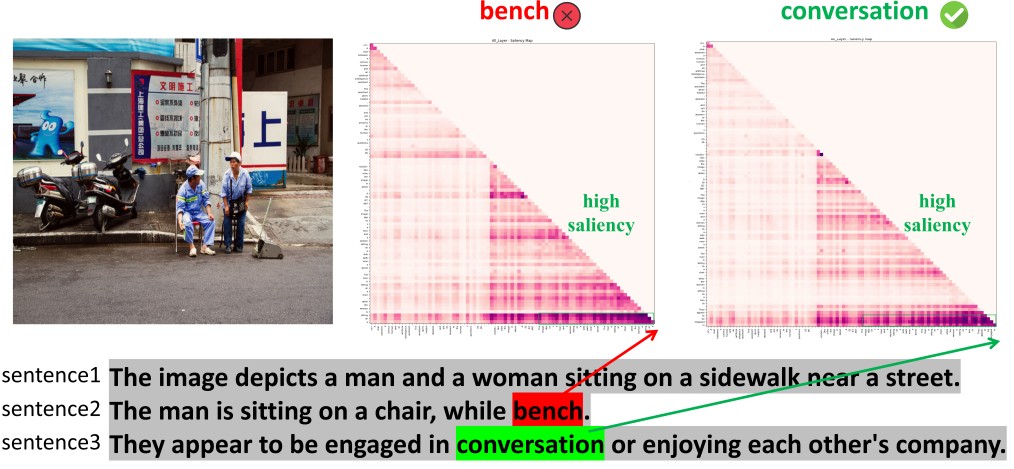

Figure 11: **Failure example**. Even though it's an hallucination token, the output saliency is still high.

Regarding our core claim that hallucinated tokens overwhelmingly exhibit low saliency, although this is strongly supported by extensive statistical evidence, we also identify several failure cases in which hallucinated tokens instead display relatively high saliency scores. Figure 11(a) illustrates such an instance: on Qwen2-VL-7B, the ground-truth answer is "a traffic cone".This contradicts the low-saliency hypothesis and reveals two fundamental limitations:

(1) Context-independent generated content: The effectiveness of the method may decrease when the content generated by the model deviates significantly from or is inconsistent with the current context. Specifically, when the saliency of a candidate token is low, indicating that the currently generated content lacks relevance to the previously generated content, SGRS will reject these tokens. However, in some cases, if the context itself is ambiguous or the input information is insufficient, the model may generate irrelevant content, which may not pass the SGRS filter even if it conforms to the rules of language generation.

(2) Some incorrect tokens may have high saliency because the model believes that the token it outputs at this time is correct (high confidence). This observation is consistent with the conclusion proposed by Adam et al. of Openai Kalai et al. (2025): "The model will make mistakes with confidence". The reason for this problem is that <1> the model is trained to output seemingly reasonable answers (high confidence) instead of expressing "I don't know". <2> after human RLHF, the model becomes overconfident.

### D.3    LONG SEQUENCE HALLUCINATION TOKEN AND LAYER EXPERIMENT

As show in Figure 12, we performed a token-level magnified visualization of the following hallucination case:

(1)**Long sequence experiments**: As shown in Figure 12, in the generated sequence, a hallucination token (e.g., "few") appears in the third sentence, while the first sentence (e.g., "preparing") and the fourth sentence (e.g., "significant") are both correct outputs. This shows that even in different sentences and adjacent positions, the saliency of hallucination tokens is significantly lower than that of the correct tokens preceding and following them.

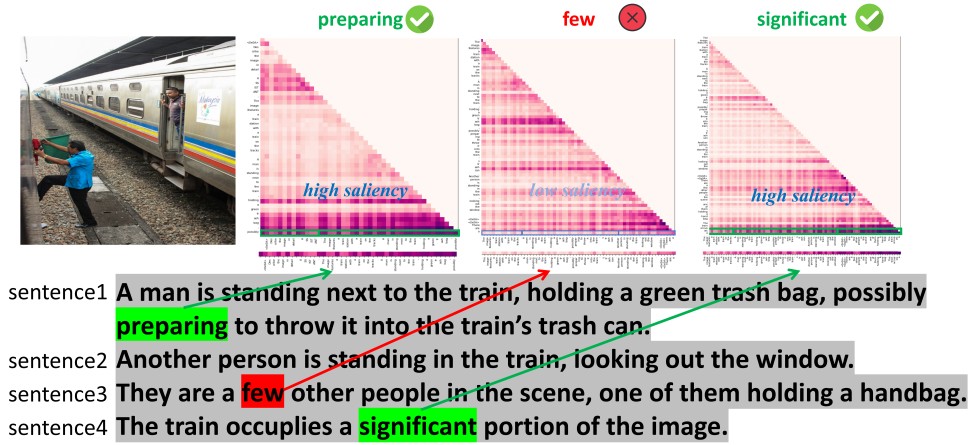

Figure 12: **Long sequence example**. A comparison of the saliency of the correct tokens before and after the hallucination token shows that the saliency of the correct tokens before and after the hallucination token is still greater than that of the original token.

(2) **Layer experiment**:As shown in Figure 13 and Figure 14, we show the saliency distribution of correct and hallucination tokens across different layers. We observe that the saliency of correct tokens is relatively high across both shallow and deep layers, while the saliency of hallucination tokens is relatively low across all layers. Regarding attention heads, they serve only as intermediate computational units and are not individually dependent. The LVLMs-Saliency calculation logic in our paper is as follows: first, calculate the saliency matrix for each attention head (Equation 4), then average it across all heads (Equation 5), finally outputting the layer-normalized saliency. Our entire method does not bind to any specific attention head, nor does it claim that any particular head/type of head is key to the connection between saliency and hallucination. We only use attention heads as the basic carrier of attention weights, and after aggregation, they are no longer considered individually.

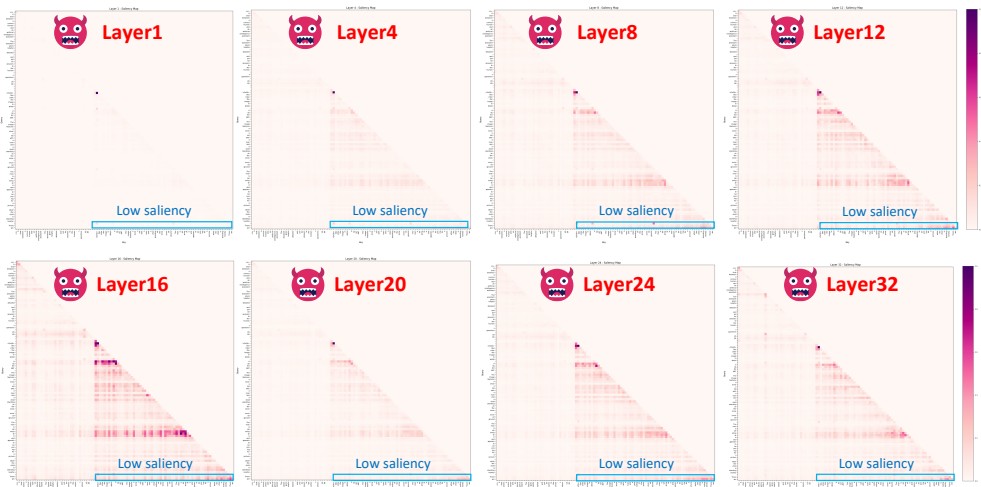

Figure 13: Saliency map of LLaVA1.5 from layer1 to layer32 (hallucination pattern).

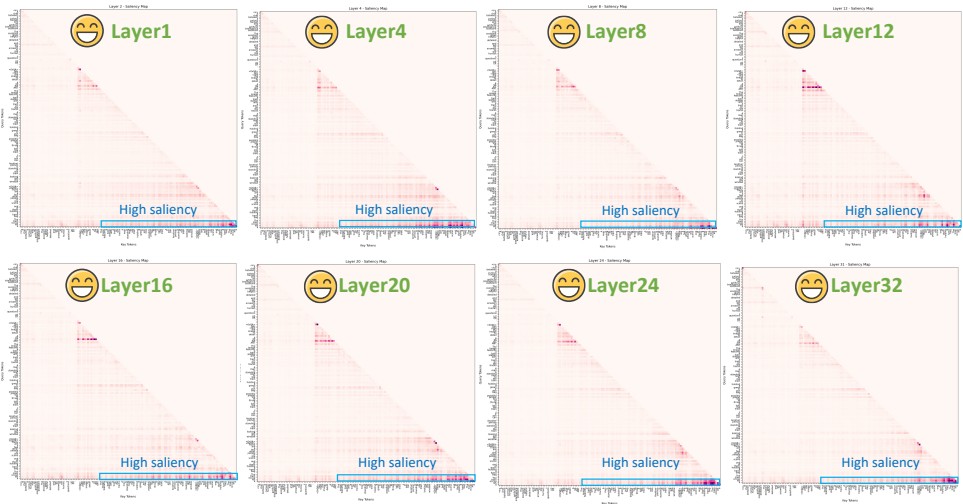

Figure 14: Saliency map of LLaVA1.5 from layer1 to layer32 (correct pattern).

