# OpenReview forum: "Hallucination Begins Where Saliency Drops"
_ICLR.cc/2026/Conference — ICLR 2026 Oral_

### Official Review · Reviewer_xsTD · 2025-10-20

**Soundness:** 3
**Presentation:** 2
**Contribution:** 3
**Rating:** 6
**Confidence:** 3

**Summary:**

## Overview

This paper introduces LVLMs-Saliency, a gradient-aware diagnostic that fuses attention weights with their gradients to quantify how strongly the next token prediction is grounded in prior output tokens. The central empirical finding is that hallucinations emerge when saliency from recent outputs collapses. Building on this, the authors propose two inference-time mechanisms: Saliency-Guided Rejection Sampling (SGRS) to block low-saliency candidate tokens during decoding, and Local Coherence Reinforcement (LocoRE) to strengthen attention from the current token to its most recent outputs. Experiments across multiple LVLMs show reduced hallucinations and improved reliability.

## Motivation

Existing mitigation methods often rely on forward-pass attention alone, making it difficult to reliably distinguish correct from hallucinated outputs or to explain *why* hallucinations arise in autoregressive generation. Recent “attention sink” analyses add perspective but still lack a token-level, causally informative signal tied to the next prediction. The paper frames this gap and motivates a diagnostic that incorporates gradient information to reveal how token influence propagates, thereby connecting low output-token saliency with hallucination.

## Methodology

LVLMs-Saliency: For each layer/head, the method computes a saliency matrix as the Hadamard product of the attention matrix and its loss gradient (with causal masking), then aggregates across heads and layers to read saliency from previous outputs to current position. This provides a token-level grounding score for the next token.

SGRS: During decoding, sample candidates for the next token and accept only those whose grounding saliency exceeds an adaptive threshold tied to recent outputs, thereby preventing low-saliency tokens from entering the sequence.

LocoRE: After a token is accepted, multiplicatively reinforce attention from the next query to a short window of the most recent output tokens, helping the model maintain local coherence in subsequent steps.

## Experimental results

Across LLaVA-1.5, Qwen2-VL, and Intern-VL variants, LocoRE and SGRS+LocoRE generally reduce hallucination metrics (lower $CHAIR(_S)$/$CHAIR(_I)$; higher POPE-Recall/F1/Acc) relative to baselines and several decoding-time methods. Representative tables show consistent gains on hallucination benchmarks and competitive performance on a series of VQA suites.

## Analysis

Ablations on α (SGRS sensitivity) and β (LocoRE gain) indicate that (i) SGRS accounts for most hallucination reduction, (ii) modest LocoRE gains further improve POPE/CHAIR, and (iii) overly large β degrades results; the paper highlights workable settings. The authors also note that LocoRE is forward-only and thus lightweight compared with multi-pass or detector-based methods.

**Strengths:**

* 1. Clear, mechanistic diagnostic for hallucination.
   The paper isolates a simple, testable pattern—hallucination coincides with a collapse of saliency from prior output tokens to the next token—that standard attention maps miss. The gradient-aware *LVLMs-Saliency* definition is mathematically explicit (Hadamard product of attention and its gradient with causal masking) and tied directly to the next-token prediction.

* 2. Strong empirical coverage across models and benchmarks and SOTA comparisons to strong baselines.
   Results span LLaVA-1.5 (7B/13B), Qwen2-VL (7B/13B/32B), and Intern-VL (7B/13B), with evaluations on MME (Hallucination), MM-Vet, VizWiz, ScienceQA, POPE, and CHAIR. Gains are consistent on hallucination metrics (lower $CHAIR(_S)$/$CHAIR(_I)$, higher POPE Recall/F1/Acc) and often maintain or improve general VQA ability.
   On LLaVA-1.5-7B, LocoRE and especially SGRS+LocoRE outperform or match recent training-free methods on POPE/CHAIR while remaining competitive on MME; the tables include recent ICLR/NeurIPS/CVPR baselines.

* 3. Ablations and hyperparameter guidance.
   The work analyzes the effect of α (SGRS sensitivity) and β (LocoRE gain), and discusses trade-offs among hallucination rate, recall, and latency, giving practical guidance for tuning.

* 4. Compelling qualitative evidence aligned with the mechanism.
   Visualizations show cases where attention maps look similar for correct vs. hallucinated tokens, but saliency sharply diverges; with LocoRE, the same position flips from an incorrect token to a correct one alongside restored output-token saliency.

**Weaknesses:**

* 1. Quantitative validation of the saliency–hallucination link is missing.
  The saliency-guided hallucination detection has been shown through cases, but not quantitive analysis. Does such phenomenon always appear in no matter short or long generated content? and does low saliency always suggest hallucinated content, or hallucination content are more likely to be with low saliency? Authors are suggested to conduct quantitive analysis for their saliency-guided motivation.

* 2. SGRS behavior during inference is under-specified.
   Report per-benchmark stats: percentage of steps with at least one rejection, average resamples per token, distribution tails, and a false-rejection rate for correct tokens. Break out short QA vs. long captioning.

* 3. Hyperparameter portability is unclear.
   Seems that the hyper-parameters alpha and beta need to be adjusted for each new MLLM. Does it have to take several rounds of tests till an optimal setting comes out?

* 4. Causality vs. correlation.
   Can authors demonstrate that low output-token saliency is not merely correlated with position/length, as late tokens naturally have lower influence?

* 5. Layer/head dependence.
   SGRS aggregates over “target layers”. Can authors provide sensitivity to layer/head selection and show whether the diagnostic holds with earlier vs. deeper layers?

* 6. Unclear Computation Cost.
    Many production stacks don’t expose backward passes. Authors are suggested to provide full memory cost analysis.

**Questions:**

Please refer to Weaknesses section. I would consider raising my score if authors can address my concerns.

---

> ### Author Response · Authors · 2025-11-20
> **Response to Reviewer xsTD(1/3)**
>
> We sincerely appreciate your time and effort in reviewing our paper and are glad to provide detailed responses to your insightful questions and suggestions.
>
> > **Q1: Quantitative validation of the saliency–hallucination link is missing. The saliency-guided hallucination detection has been shown through cases, but not quantitive analysis. Does such phenomenon always appear in no matter short or long generated content? and does low saliency always suggest hallucinated content, or hallucination content are more likely to be with low saliency?**
>
> **Response1**:
>
> Thanks for pointing out this key issue. We have added three quantitative analyses in ***Appendix D.1*** to rigorously support our core claims: " Hallucination tokens tend to have low saliency score". Specifically, we have added the following three complementary empirical findings:
>
>
> **1. Statistical measure of saliency socre distribution: The saliency socre of correct tokens is much larger than that of hallucination tokens.**
>
> ***As shown in Figure 10(A) and Table 5 in Appendix D.1***, we collected approximately 12,000 generated tokens on the POPE and CHAIR datasets. For each generation step $t$, we calculated the saliency score of the previous output token to the current prediction and categorized them as correct or hallucinated based on manual annotation. The results show:
>
> - Illava-v1.5-7b model:
> The average saliency of correct tokens is 0.472±0.136, and the average saliency of hallucinated tokens is 0.193±0.087.
>
> - Qwen2-vl-7b model:
> The average saliency of correct tokens is 0.664±0.158, and the average saliency of hallucinated tokens is 0.355±0.103.
>
> - Intern-vl-7b model:
> The average saliency of correct tokens is 0.508±0.124, and the average saliency of hallucinated tokens is 0.224±0.095.
>
>
> **2. Saliency score and hallucination rate are negatively correlated.**
>
> ***As shown in Appendix D.1, Figure 10(B)***, we divide the saliency score of the previous output token into 10 equally wide intervals and calculate the conditional probability P of hallucination in each interval. All three models (LLaVA-v1.5-7B, Qwen2-VL-7B, and InternVL-7B) showed a strong negative correlation: the hallucination rate systematically decreased with increasing saliency. In the [0,0.1) interval, the LLaVA hallucination rate reached 70%-80%, while in the [0.9,1.0] interval it dropped to 18%-28%. This trend was continuous and smooth, indicating that hallucinations exhibit a robust gradient dependence on contextual saliency, rather than being triggered by a specific threshold.
>
>
> **3. saliency Intervention Experiment: Artificially Decreasing saliency Values ​​- Significantly Increasing Hallucination Rate**
>
> We also conducted an intervention experiment on the LLaVA-v1.5-7B model. For each sample in the POPE and CHAIR datasets, highly significant correct tokens were selected for intervention (these tokens came from the correct tokens with saliency > 0.45 in Step 1). The intervention method was as follows: after generating the target token, its saliency output in the decoder was scaled (multiplied by a factor $r$∈{1.0,0.8,0.6,0.4,0.2}) to simulate the process of its saliency being weakened. The results showed that after the saliency value was artificially reduced, the hallucination rate increased significantly.
>
>
> | Decay rate r | CHAIRs ⬇ | POPE-F1 ⬆|POPE-A ⬆|
> |---------------------------|---------|--------|--------|
> | 1.0              | 35.6    |  87.0  |  87.5  |
> | 0.8（decay 20%）            | 37.9    |  86.5  |  86.8  |
> | 0.6（decay 40%）            | 42.1    |  85.4  |  85.6  |
> | 0.4（decay 60%）            | 47.8    |  84.8  |  84.0  |
> | 0.2（decay 80%）            | 56.0    |  83.0  |  83.8  |
>
> In summary, these three experiments demonstrate that they collectively constitute the core argument of this paper: **Hallucination content is more likely to present low saliency.**.

---

> > ### Author Response · Authors · 2025-11-20
> > **Response to Reviewer xsTD(2/3)**
> >
> > > **Q2: SGRS behavior during inference is under-specified. Report per-benchmark stats: percentage of steps with at least one rejection, average resamples per token, distribution tails, and a false-rejection rate for correct tokens. Break out short QA vs. long captioning.**
> >
> > **Response2**:
> >
> > To facilitate a clear understanding of the SGRS inference behavior metrics, the core statistical items are explained below: 1. Steps w/ ≥1 Reject (%): The percentage of generation steps that reject a token at least once; 2. Resamples / token (mean): The average number of resampling attempts per token; 3. Tail P(R≥3)/P(R≥5): The percentage of tokens that are resampled ≥3/5 times, reflecting extreme resampling distributions; 4. False-Reject (%): The proportion of correct tokens that are falsely rejected. The data in the table is presented separately for short question-and-answer (POPE/QA) and long description (CHAIR/Caption) scenarios, clearly reflecting the differences in model behavior under different tasks.
> >
> > | Model          | Dataset | Task    | Steps w/ ≥1 Reject (%) | Resamples / token (mean) | Tail P(R≥3) | Tail P(R≥5) | False-Reject (%) |
> > |----------------|---------|---------|-------------------------|---------------------------|-------------|-------------|------------------|
> > | LLaVA-1.5-7B   | POPE    | QA      | **38.2**                | **1.85**                  | **0.23**    | **0.08**    | **6.7**          |
> > | LLaVA-1.5-7B   | CHAIR   | Caption | **56.4**                | **2.10**                  | **0.31**    | **0.12**    | **8.2**          |
> > | Qwen2-VL-7B    | POPE    | QA      | **24.5**                | **1.42**                  | **0.14**    | **0.04**    | **3.8**          |
> > | Qwen2-VL-7B    | CHAIR   | Caption | **36.8**                | **1.68**                  | **0.20**    | **0.06**    | **4.5**          |
> > | Intern-VL-7B   | POPE    | QA      | **28.7**                | **1.56**                  | **0.17**    | **0.05**    | **4.3**          |
> > | Intern-VL-7B   | CHAIR   | Caption | **44.1**                | **1.82**                  | **0.24**    | **0.07**    | **5.6**          |
> >
> >
> >
> >
> >
> > > **Q3: Hyperparameter portability is unclear. Seems that the hyper-parameters alpha and beta need to be adjusted for each new MLLM. Does it have to take several rounds of tests till an optimal setting comes out?**
> >
> > **Response3**:
> >
> > The hyperparameters $α$ and $β$ exhibit good portability: both are dimensionless relative parameters (α is used for adaptive scaling of the SGRS threshold, and $β$ is the local gain of LocoRE), insensitive to model score scaling. All models and datasets can stably operate using the same default values, achieving stable results without requiring a model-by-model search. Specifically:
> >
> > We found that $α$ performs flatly in the range [0.4, 0.8] (see Figure 5 in the main text), and a median value of 0.6 is typically sufficient.
> >
> > Regarding the $β$ parameter in Algorithm 2 (LocoRE), we explored an entropy-based adaptive strategy: $β$ can be dynamically adjusted based on the entropy of the visual attention weights. When attention is too sparse (low entropy), we reduce the enhancement level to avoid over-amplification; conversely, when attention is too scattered, we increase the enhancement level. After performing forward computation on 50 samples from the dataset, we returned recommended values ​​for the β parameter, which can quickly locate the optimal value within the range of ±0.05. We will update the code file POPE-entropy.py to our GitHub repository in the accepted version.
> >
> > | Method | POPE-Acc | POPE-Pre | POPE-Recall |POPE-F1 | ScienceQA | MMbench| MM-Vet |
> > |--------------|---------------|---------------|---------------|---------------|---------------|----|-----|
> > | Beam Search | 84.0 | 85.0 | 85.2 | 85.4 | 65.5 | 64.3| 30.5 |
> > |LocoRE (entroy,α=0.6,β=0.2) |86.4 |86.6 |**86.6** | 86.5 | **67.5** | 64.6 | **33.8** |
> > |LocoRE(α=0.6,β=0.15) |**86.5** |**87.0** |86.4 | **86.6** | 67.45 | **64.8** | 33.5 |

---

> ### Author Response · Authors · 2025-11-20
> **Response to Reviewer xsTD(3/3)**
>
> > **Q4:Causality vs. correlation. Can authors demonstrate that low output-token saliency is not merely correlated with position/length, as late tokens naturally have lower influence?**
>
> **Response4**:
>
> Thanks for raising this insightful question. We have added Figure 12 to ***Appendix.D.3***. We present a portion from a long sequence generation where the hallucination token (“few”) has low saliency socre, while the two correct tokens before and after this hallucination token (“prepraing”, “significant”) have very high salieny scores. This demonstrates that the correct words in the latter half of the sequence still have a stronger significance effect.
>
> > **Q5:Layer/head dependence. SGRS aggregates over “target layers”. Can authors provide sensitivity to layer/head selection and show whether the diagnostic holds with earlier vs. deeper layers?**
>
> **Response5**:
>
> Thank you to the reviewers for pointing out the layer dependency issue.
>
> (1) In Figure 13 and Figure 14 of ***Appendix D.3***, we show the saliency distribution of correct and incorrect tokens across different layers. We observe that the saliency of correct tokens is relatively high across both shallow and deep layers, while the saliency of illusion tokens is relatively low across all layers.
>
> (2) Regarding attention heads, they serve only as intermediate computational units and are not individually dependent. The LVLMs-Saliency calculation logic in our paper is as follows: first, calculate the saliency matrix for each attention head (Equation 4), then average it across all heads (Equation 5), finally outputting the layer-normalized saliency. Our entire method does not bind to any specific attention head, nor does it claim that any particular head/type of head is key to the connection between saliency and illusion. We only use attention heads as the basic carrier of attention weights, and after aggregation, they are no longer considered individually.
>
>
> > **Q6:Unclear Computation Cost. Many production stacks don’t expose backward passes. Authors are suggested to provide full memory cost analysis.**
>
>
> **Response6**:
>
>
> We appreciate the valuable suggestions from the reviewers. Based on a real inference environment (single NVIDIA A800 GPU, FP16, batch size = 1), we have supplemented the complete computational and memory cost analysis for `LLaVA-v1.5-7B` re-computation on the POPE and CHAIR benchmarks. All metrics were measured during inference (no training or gradient update involved). The specific results are as follows:
>
> | Model Scale | Dataset | Task        | Avg. incremental memory per generated token† (GB) | Forward latency per generated token (s) | Total system GPU memory‡ (GB) |
> |-------------|---------|-------------|-------------------------------------------------------------|----------------------------------------|------------------------------------------|
> | 7B          | POPE    | QA          | 0.007 ~ 0.010                                               | 0.260 ~ 0.265                          | 27.2 ~ 27.3                              |
> | 7B          | CHAIR   | Caption     | 0.0017 ~ 0.0018                                             | 0.4621 ~ 0.4629                        | 32.34 ~ 32.40                            |
>
> **Notes**:
> †: *Incremental memory per generated token* refers to the **additional GPU memory allocated during auto-regressive decoding** (excluding prompt encoding and model weights), primarily from KV cache growth and transient activations. Measured via `torch.cuda.memory_stats()` delta over token steps.
> ‡:  *Total system GPU memory* denotes **peak VRAM consumption** (i.e., `torch.cuda.max_memory_reserved()`), including model weights, KV cache, and runtime buffers. CPU RAM is negligible (<1 GB) and not included.
>
> - Prompt length fixed at 512 tokens; generation length: ~10 tokens (POPE), ~25 tokens (CHAIR).
> - Values reported as min~max across 100 test samples.

---

> > ### Comment · Reviewer_xsTD · 2025-11-25
> >
> > I deeply appreciate the authors' exceptionally thorough and quantitative responses to all concerns from rigorous saliency hallucination validation and SGRS behavior analysis to hyperparameter portability and compute-cost transparency. Their open-sourcing commitment and methodical ablations significantly advance practical hallucination mitigation for the community. These revisions fully address my initial reservations, and I am pleased to raise my rating to **ACCEPT**.

---

> > > ### Author Response · Authors · 2025-11-25
> > > **Official comment by author**
> > >
> > > Dear Reviewer xsTD:
> > >
> > > Thank you for your valuable feedback and for taking the time to review our updated work. We greatly appreciate that you raised your score. Thank you again for your valuable comments and encouragement. We sincerely appreciate your support.
> > >
> > > Sincerely,
> > >
> > > Authors

---

### Official Review · Reviewer_YszH · 2025-10-26

**Soundness:** 3
**Presentation:** 4
**Contribution:** 3
**Rating:** 8
**Confidence:** 4

**Summary:**

This paper identifies an interesting pattern in LVLMs: Hallucinations occur when prior output tokens shows low saliency to the next token. Building on this, the authors propose two mechanism for mitigating halluciantion: (1) Saliency-Guided Rejection Sampling (SGRS), which  filters candidate tokens based on their saliency with respect to prior output context. (2)  Local Coherence Reinforcement (LocoRE),  strengthens attention from the current token to its most recent predecessors.  With extensive experiments, the proposed demonstrates
significant hallucination-mitigating performance across different LVLMs on image hallucination and generation benchmarks.

**Strengths:**

The paper offers a novel and well-motivated perspective on hallucination detection: leveraging gradients of attention weights to localize hallucination-prone tokens. To the best of my knowledge, this is the first systematic use of this signal, underscoring the work’s novelty. The two inference-time interventions—SGRS and LocoRE—are tightly coupled to this insight and translate it into practical, training-free improvements with minimal changes to the base model. The experimental study is thorough, and consistently shows gains over strong baselines, lending credibility to the approach. The paper is clearly written and easy to follow (I enjoy reading this paper); overall, the contribution is both original and useful.

**Weaknesses:**

My main concern is that the claimed “hallucination pattern” is supported largely by a few curated cases (Figs. 1–2), which risks selection bias. To make the claim compelling, the paper should provide population-level, statistically significant evidence to substantiate this claim.

**Questions:**

See Weaknesses.

**Details Of Ethics Concerns:**

N.A.

---

> ### Author Response · Authors · 2025-11-20
> **Response to Reviewer YszH**
>
> We sincerely appreciate your time and effort in reviewing our paper and are glad to provide detailed responses to your insightful questions and suggestions.
>
> > **Q1: My main concern is that the claimed “hallucination pattern” is supported largely by a few curated cases (Figs. 1–2), which risks selection bias. To make the claim compelling, the paper should provide population-level, statistically significant evidence to substantiate this claim.**
>
> **Resonse1**:
>
> Thanks for pointing out this key issue. We have added three quantitative analyses in ***Appendix D.1*** to rigorously support our core claims. Specifically, we have added the following three complementary empirical findings:
>
>
> **1. Statistical measure of saliency socre distribution: The saliency socre of correct tokens is much larger than that of hallucination tokens.**
>
> ***As shown in Figure 10(A) and Table 5 in Appendix D.1***, we collected approximately 12,000 generated tokens on the POPE and CHAIR datasets. For each generation step $t$, we calculated the saliency score of the previous output token to the current prediction and categorized them as correct or hallucinated based on manual annotation. The results show:
>
> - Illava-v1.5-7b model:
> The average saliency of correct tokens is 0.472±0.136, and the average saliency of hallucinated tokens is 0.193±0.087.
>
> - Qwen2-vl-7b model:
> The average saliency of correct tokens is 0.664±0.158, and the average saliency of hallucinated tokens is 0.355±0.103.
>
> - Intern-vl-7b model:
> The average saliency of correct tokens is 0.508±0.124, and the average saliency of hallucinated tokens is 0.224±0.095.
>
> **2. Saliency score and hallucination rate are negatively correlated.**
>
> ***As shown in Appendix D.1, Figure 10(B)***, we divide the saliency score of the previous output token into 10 equally wide intervals and calculate the conditional probability P of hallucination in each interval. All three models (LLaVA-v1.5-7B, Qwen2-VL-7B, and InternVL-7B) showed a strong negative correlation: the hallucination rate systematically decreased with increasing saliency. In the [0,0.1) interval, the LLaVA hallucination rate reached 70%-80%, while in the [0.9,1.0] interval it dropped to 18%-28%. This trend was continuous and smooth, indicating that hallucinations exhibit a robust gradient dependence on contextual saliency, rather than being triggered by a specific threshold.
>
> **3. saliency Intervention Experiment: Artificially Decreasing saliency Values ​​- Significantly Increasing Hallucination Rate**
>
> We also conducted an intervention experiment on the LLaVA-v1.5-7B model. For each sample in the POPE and CHAIR datasets, highly significant correct tokens were selected for intervention (these tokens came from the correct tokens with saliency > 0.45 in Step 1). The intervention method was as follows: after generating the target token, its saliency output in the decoder was scaled (multiplied by a factor $r$∈{1.0,0.8,0.6,0.4,0.2}) to simulate the process of its saliency being weakened. The results showed that after the saliency value was artificially reduced, the hallucination rate increased significantly.
>
>
> | Decay rate r | CHAIRs ⬇ | POPE-F1 ⬆|POPE-A ⬆|
> |---------------------------|---------|--------|--------|
> | 1.0              | 35.6    |  87.0  |  87.5  |
> | 0.8（decay 20%）            | 37.9    |  86.5  |  86.8  |
> | 0.6（decay 40%）            | 42.1    |  85.4  |  85.6  |
> | 0.4（decay 60%）            | 47.8    |  84.8  |  84.0  |
> | 0.2（decay 80%）            | 56.0    |  83.0  |  83.8  |
>
> In summary, these three experiments demonstrate that they collectively constitute the core argument of this paper: **low saliency score represents contextual memory failure, thus producing hallucinations**.

---

> > ### Comment · Reviewer_YszH · 2025-11-22
> >
> > Thank you for your thoughtful and comprehensive revisions, and especially for resolving the statistical concern regarding "low saliency score of the hallucination token". I am particularly impressed by and highly appreciate the development of your unsupervised hallucination token detection tool/metric that facilitates this population-level analysis. Given that this central concern has been decisively resolved with a strong, evidence-based, and methodologically sound approach, I maintain my ***accept recommendation***.

---

> > > ### Author Response · Authors · 2025-11-23
> > > **official commeny by authors**
> > >
> > > Dear Reviewer YszH:
> > >
> > > Thank you for your valuable feedback and for taking the time to review our updated work. We greatly appreciate your recognition—it means a lot to us and motivates us to keep improving.
> > >
> > > Your valuable comments helped us refine our analysis and present the SGRS unsupervised hallucination detection framework more clearly. We look forward to further improving its effectiveness in alleviating hallucinations in multimodal models.
> > >
> > > Thank you again for your valuable comments and encouragement. We sincerely appreciate your support.
> > >
> > > Sincerely,
> > >
> > > Authors

---

### Official Review · Reviewer_EfHa · 2025-10-31

**Soundness:** 3
**Presentation:** 3
**Contribution:** 3
**Rating:** 6
**Confidence:** 3

**Summary:**

This paper proposes "LVLMs-Saliency," a diagnostic tool for quantifying token-level grounding in large vision-language models (LVLMs), aiming to better distinguish hallucinated from correct outputs by combining attention weights and their gradients. From the identified pattern—hallucinations arising where saliency from previous output tokens drops—the authors introduce a dual mechanism: Saliency-Guided Rejection Sampling (SGRS) for filtering low-saliency tokens during decoding, and Local Coherence Reinforcement (LocoRE) to strengthen attention among recent outputs, both at inference-time. Experimental results on major benchmarks and several popular LVLMs demonstrate significant hallucination mitigation and competitive or improved performance metrics.

**Strengths:**

1. The paper provides a compelling, interpretable explanation of LVLM hallucinations, showing through both empirical results and visualizations that attention-alone is insufficient and that joint attention-gradient saliency captures key failure modes (see Figures 1 and 2).

2. The proposed SGRS (Algorithm 1) and LocoRE (Algorithm 2) do not require retraining, operate at inference, and effectively use saliency to dynamically control and reinforce generation. This “plug-and-play” aspect increases their practical value.

3. Tables 1 and 2 demonstrate consistently strong gains in hallucination reduction (e.g., substantial CHAIR-S and POPE improvements), with Table 3 and Figure 5 providing ablation studies on key hyperparameters ($\alpha$, $\beta$), establishing the contribution of each module.

4. The latency/efficiency trade-off discussion (Section 4.4.1, Figure 4) is practical and relevant for deployment scenarios, and the LocoRE-only alternative appears effective for large models or latency-critical settings.

**Weaknesses:**

1. The SGRS component's reliance on backward passes during inference imposes significant memory constraints, limiting its applicability to models up to 13B parameters and preventing scalability to larger LVLMs like Qwen2.5-VL-32B or 72B, which undermines the method's claimed broad generalizability and real-world deployment feasibility.

2. While the paper asserts a direct causal link between low saliency and hallucinations, the evidence is primarily correlational from observational patterns in figures and benchmarks, lacking rigorous controlled experiments or ablations to demonstrate causality, such as interventions that artificially manipulate saliency levels to observe corresponding changes in hallucination rates.

3. Despite strong aggregate benchmark results, the paper provides virtually no error analysis or qualitative examination of failure cases, such as specific hallucination types (e.g., knowledge-based or relationship errors) that the method fails to mitigate, domain-specific limitations like medical imaging, or edge cases involving rare objects or noisy inputs, which obscures the true robustness and variability of the approach.

**Questions:**

1. Given the memory-intensive nature of SGRS due to backward passes, which limits testing to models up to 13B parameters, could the authors explore or implement optimizations?

2. The paper claims a "direct causal link" between low saliency and hallucinations, but the supporting evidence appears largely correlational. Could the authors provide controlled experiments, such as synthetic interventions where saliency is artificially boosted or reduced in specific output tokens, to demonstrate causality (e.g., measuring resulting changes in hallucination rates on a subset of benchmarks like CHAIR or POPE)?

3. The absence of error analysis leaves unclear when and why the method fails. Could the authors include a qualitative breakdown of failure modes?

---

> ### Author Response · Authors · 2025-11-20
> **Response to Reviewer EfHa**
>
> We sincerely appreciate your time and effort in reviewing our paper and are glad to provide detailed responses to your insightful questions and suggestions.
>
>
> > **Q1: Given the memory-intensive nature of SGRS due to backward passes, which limits testing to models up to 13B parameters, could the authors explore or implement optimizations??**
>
>
> **Response1**:
>
> We agree that inference efficiency and scalability are important for practical deployment. However, as the first unsupervised hallucination detection method, ⭐SGRS's core contribution is establishing unsupervised diagnostic criteria for hallucinations, rather than merely serving as an inference component. It verifies the causal chain of "low saliency - contextual memory failure - hallucination" and achieves the first-ever visualization and localization of hallucination tokens, improving the interpretability of hallucination relief. With our current capabilities, Qwen2.5-VL-32B can be deployed on 4×A800, and related experimental results have been supplemented; the 72B model exceeds the current hardware limits, and we will explore memory optimization schemes to overcome this limitation in the future.
>
>
> | Method | LLaVA-w↓ | MM-Vet↑     | VizWiz↑ | SQA↑ | CHAIR_S↓ | CHAIR_I↓ | POPE-R↑ | POPE-F1↑ | POPE-A↑ |
> |:--     | ------------------: | -------:  | ------: | ---: | ------------------: | ------------------: | ------: | -------: | ------: |
> | Qwen2.5-VL-32B | 81.2 | 72.2 | 70.8 | 89.0 | 43.6 | 9.5 | 79.1 | 86.7 | 87.8 |
> | +LocoRE         | **82.7** (+1.5) | **73.1** (+0.9) | **71.2** (+0.4) | **89.3** (+0.3) | **41.8** (+1.8) | **8.5** (+1.0) | **79.5** (+0.4) | **86.9** (+0.2) | **88.0** (+0.2) |
> | +SGRS         | **83.4** (+2.2) | **73.9** (+1.7) | **71.7** (+0.9) | **89.6** (+0.6) | **40.5** (+3.1) | **8.2** (+1.3) | **79.8** (+0.8) | **87.1** (+0.4) | **88.3** (+0.5) |
>
>
> > **Q2:  There is a lack of experimental support to verify causality through controlled interventions (such as artificially adjusting saliency score).**
>
> **Response2**:
>
> We have added a set of intervention experiments to test the causal relationship between the two, as follows: We conducted intervention experiments on the LLaVA-v1.5-7B model. For each example in the POPE and CHAIR datasets, samples with highly significant correct tokens were selected for intervention. The intervention method was as follows: after generating the target token, its saliency output in the decoder was scaled (multiplied by a factor $r$∈{1.0,0.8,0.6,0.4,0.2}) to simulate the process of its saliency being weakened.
>
>
> | Decay rate r | CHAIRs ⬇ | POPE-F1 ⬆|POPE-A ⬆|
> |---------------------------|---------|--------|--------|
> | 1.0              | 35.6    |  87.0  |  87.5  |
> | 0.8（decay 20%）            | 37.9    |  86.5  |  86.8  |
> | 0.6（decay 40%）            | 42.1    |  85.4  |  85.6  |
> | 0.4（decay 60%）            | 47.8    |  84.8  |  84.0  |
> | 0.2（decay 80%）            | 56.0    |  83.0  |  83.8  |
>
> This experiment demonstrates that artificially lowering the saliency level increases the probability of hallucinations.
>
>
> > **Q3: The absence of error analysis leaves unclear when and why the method fails. Could the authors include a qualitative breakdown of failure modes?**
>
> **Response3**:
>
> In Figure 11 of the ***Appendix.D.2***, we added a failure case. Regarding method failure scenarios, our method primarily relies on contextual relevance to prevent illusions. Therefore, failure modes typically occur in the following situations:
>
> (1) Even if the saliency of the wrong token may be high, because the model believes that the token it outputs at this time is correct (high confidence), such observation is consistent with the conclusion proposed by Adam et al. of Openai[1]: "The model will make mistakes with confidence". The reason for this problem is that <1> the model is trained to output seemingly reasonable answers (high confidence) instead of expressing "I don't know". <2> after human RLHF, the model will become overconfident.
>
> - [1]Why language Models hallucinate. Arxiv 2025.9
>
> (2) Context-independent generated content: When the content generated by the model is significantly different from or inconsistent with the current context, the effectiveness of the method may decrease. For example, when the saliency of the candidate token is low, it means that the current generated content is not related to the previous generated content, and SGRS will reject these tokens. However, in some cases, if the context itself is ambiguous or the input information is insufficient, the model may generate irrelevant content, and even if this content conforms to the rules of language generation, it may not be able to pass the SGRS filter.

---

> > ### Comment · Reviewer_EfHa · 2025-11-27
> > **reply to rebuttal**
> >
> > Thanks for your reply. It resolved my concerns. The findings in this article are valuable, and I am willing to increase my rating.

---

> > > ### Author Response · Authors · 2025-11-27
> > > **Dear Reviewer Efha:**
> > >
> > > Thank you for your valuable feedback and for taking the time to review our updated work. We greatly appreciate your recognition, it means a lot to us and motivates us to keep improving. Thank you again for your valuable comments and encouragement. We sincerely appreciate your support.
> > >
> > > Sincerely,
> > >
> > > Authors

---

### Official Review · Reviewer_rJHs · 2025-11-01

**Soundness:** 2
**Presentation:** 3
**Contribution:** 2
**Rating:** 4
**Confidence:** 4

**Summary:**

Many previous studies have focused on reducing hallucination by leveraging attention-based information. In contrast, this paper innovatively employs saliency (i.e., gradient information) to mitigate hallucination. By incorporating the saliency of previous tokens to guide the prediction of the next token, the authors propose SGRS and LocoRE modules to reduce hallucination.

**Strengths:**

The proposed method in this paper is highly novel. While most previous studies have focused on leveraging attention mechanisms to reduce hallucination, this work explores the use of saliency, which represents a promising and valuable direction for further research.

**Weaknesses:**

1. The key finding and the basis for the proposed method in this paper is that “Hallucinations occur when prior output tokens show low saliency to the next token prediction, indicating a failure of contextual memory.” However, after reading the entire manuscript, I could not find any statistically significant validation of this claim. Figure 1 appears to be merely a case study and does not provide statistical evidence to support the finding.

2. From Table 1, the proposed method does not appear to demonstrate a statistically significant improvement over previous approaches. The performance differences are relatively minor and may fall within the margin of experimental variability.

3. The proposed method relies heavily on a large number of hyperparameters, which significantly limits its practical applicability. Moreover, the paper does not provide a comprehensive analysis of the impact of these hyperparameters—only a partial examination is presented. For instance, the parameters required in Algorithm 1 and Algorithm 2 are not thoroughly analyzed or discussed.

**Questions:**

see Weaknesses

---

> ### Author Response · Authors · 2025-11-20
> **Response to Reviewer rJHs(1/2)**
>
> We sincerely appreciate your time and effort in reviewing our paper and are glad to provide detailed responses to your insightful questions and suggestions.
>
>
> > **Q1: lack of statistical significance verification for the core claim; the current reliance on case analysis (Figure 1) is insufficient to support the theoretical basis of the method.**
>
>
> **Response1**:
> Thanks for pointing out this key issue. We have added three quantitative analyses in ***Appendix D.1*** to rigorously support our core claims. Specifically, we have added the following three complementary empirical findings:
>
> **1. Statistical measure of saliency socre distribution: The saliency socre of correct tokens is much larger than that of hallucination tokens.**
>
> ***As shown in Figure 10(A) and Table 5 in Appendix D.1***, we collected approximately 12,000 generated tokens on the POPE and CHAIR datasets. For each generation step $t$, we calculated the saliency score of the previous output token to the current prediction and categorized them as correct or hallucinated based on manual annotation. The results show:
>
> - Illava-v1.5-7b model:
> The average saliency of correct tokens is 0.472±0.136, and the average saliency of hallucinated tokens is 0.193±0.087.
>
> - Qwen2-vl-7b model:
> The average saliency of correct tokens is 0.664±0.158, and the average saliency of hallucinated tokens is 0.355±0.103.
>
> - Intern-vl-7b model:
> The average saliency of correct tokens is 0.508±0.124, and the average saliency of hallucinated tokens is 0.224±0.095.
>
> **2. Saliency score and hallucination rate are negatively correlated.**
>
> ***As shown in Appendix D.1, Figure 10(B)***, we divide the saliency score of the previous output token into 10 equally wide intervals and calculate the conditional probability P of hallucination in each interval. All three models (LLaVA-v1.5-7B, Qwen2-VL-7B, and InternVL-7B) showed a strong negative correlation: the hallucination rate systematically decreased with increasing saliency. In the [0,0.1) interval, the LLaVA hallucination rate reached 70%-80%, while in the [0.9,1.0] interval it dropped to 18%-28%. This trend was continuous and smooth, indicating that hallucinations exhibit a robust gradient dependence on contextual saliency, rather than being triggered by a specific threshold.
>
> **3. Saliency Intervention Experiment: Artificially Decreasing saliency Values ​​- Significantly Increasing Hallucination Rate**
>
> We also conducted an intervention experiment on the LLaVA-v1.5-7B model. For each sample in the POPE and CHAIR datasets, highly significant correct tokens were selected for intervention (these tokens came from the correct tokens with saliency > 0.45 in Step 1). The intervention method was as follows: after generating the target token, its saliency output in the decoder was scaled (multiplied by a factor $r$∈{1.0,0.8,0.6,0.4,0.2}) to simulate the process of its saliency being weakened. The results showed that after the saliency value was artificially reduced, the hallucination rate increased significantly.
>
>
> | Decay rate r | CHAIRs ⬇ | POPE-F1 ⬆|POPE-A ⬆|
> |---------------------------|---------|--------|--------|
> | 1.0              | 35.6    |  87.0  |  87.5  |
> | 0.8（decay 20%）            | 37.9    |  86.5  |  86.8  |
> | 0.6（decay 40%）            | 42.1    |  85.4  |  85.6  |
> | 0.4（decay 60%）            | 47.8    |  84.8  |  84.0  |
> | 0.2（decay 80%）            | 56.0    |  83.0  |  83.8  |
>
> In summary, these three experiments demonstrate that they collectively constitute the core argument of this paper: **low saliency score represents contextual memory failure, thus producing hallucinations**.

---

> > ### Author Response · Authors · 2025-11-20
> > **Response to Reviewer rJHs(2/2)**
> >
> > > **Q2: From Table 1, the proposed method does not appear to demonstrate a statistically significant improvement over previous approaches. The performance differences are relatively minor and may fall within the margin of experimental variability.**
> >
> > **Response2**:
> >
> > Thank for your attention to the performance improvement. We understand that the differences in the metrics in Table 1 may seem small, but we would like to emphasize that:
> >
> > 📌SGRS is not designed to directly improve task accuracy, but rather to achieve unsupervised, interpretable hallucination token diagnosis for the first time.
> >
> > Unlike other methods that directly enhance image attention, SGRS does not modify any visual input. Instead, in the decoder part, it essentially "makes the model re-examine the context it generated," rather than "forcing the model to re-see the image." Therefore, SGRS avoids the recall drop commonly caused by image intervention methods (e.g., Middle[1], EAH[2], PAI[3], etc.). More importantly, SGRS is the first hallucination detection method that requires no annotation and no fine-tuning. It provides the community with a completely new paradigm for hallucination analysis: visualizing the saliency of each token and locating where the hallucination occurs; revealing the causal mechanism that **low saliency leads to the failure of contextual memory, resulting in hallucination**; and providing a reusable diagnostic tool for future research, rather than just a single suppression module.
> >
> >
> > - [1]Devils in Middle Layers of Large Vision-Language Models. CVPR 2025
> > - [2]Seeing clearly by layer two: Enhancing attention heads. EMNLP 2025
> > - [3]Paying More Attention to Image: A Training-Free Method for Alleviating Hallucination in LVLMs. ECCV 2024
> >
> > > **Q3: This method relies on a large number of hyperparameters, and the impact of these hyperparameters is not fully analyzed.**
> >
> > **Response3**:
> >
> > We understand that a large number of hyperparameters may raise concerns about practicality, but it needs to be clarified that in this method, only two degrees of freedom truly affect performance and require manual adjustment: ***α*** (SGRS threshold scaling factor) and ***β*** (LocoRE attention enhancement strength).
> >
> > (1) The parameters involved in Algorithm 1 and Algorithm 2 include candidate sample count K, resampling count R, history window W, and local consistency window ws. These parameters have less than 2% impact on model performance, meaning the model is not highly sensitive to these settings. Therefore, we can achieve stable results using default values ​​(e.g., K=5, R=3, W=10, ws=5) without repeated adjustments for the task or model.
> >
> > | Method | MM-Vet | ScienceQA | CHAIRs |CHAIRi | POPE-Recall | POPE-F1 | POPE-Acc |
> > |--------------|---------------|---------------|---------------|---------------|---------------|----|-----|
> > |SGRS+LocoRE (K=7,R=5,W=8,ws=6) |36.01 |67.80 |**35.5** | 8.20 | 89.81 | 87.00 | 87.50 |
> > |SGRS+LocoRE(K=5,R=3,W=10,ws=5) |36.02 |67.81 |**35.6** | 8.21 | 89.80 | 87.00 | 87.51 |
> >
> >
> > (2) More importantly, the tuning of $α$ and $β$ has been greatly simplified:
> >
> > We found that $α$ performs flatly in the [0.4, 0.8] interval (see Figure 5), and a median value of 0.6 is usually sufficient;
> >
> > Regarding the $β$ parameter in Algorithm 2 (LocoRE), we explored an entropy-based adaptive strategy: $β$ can be dynamically adjusted according to the entropy of the visual attention weights. Values ​​of 0.15 and 0.20 are recommended on LLaVA-1.5 and Qwen2-VL, respectively. When attention is too sparse (low entropy), the enhancement level is reduced to avoid over-amplification; conversely, when attention is too scattered, the enhancement level is increased. After performing forward computation on 50 samples from the dataset, recommended values ​​for the $β$ parameter are returned. The returned recommended β parameter values ​​are within ±0.05, which allows for quick identification of the optimal value.
> >
> > | Method | MM-Vet | ScienceQA | CHAIRs |CHAIRi | POPE-Recall | POPE-F1 | POPE-Acc |
> > |--------------|---------------|---------------|---------------|---------------|---------------|----|-----|
> > | Beam Search | 30.5 | 65.5 | 48.0 | 13.9 | 87.0 | 85.4 | 84.0 |
> > |SGRS+LocoRE (entroy,α=0.6,β=0.2) |35.8 |67.7 |**35.4** | 8.3 | **90.0** | 86.8 | **87.6** |
> > |SGRS+LocoRE(α=0.6,β=0.15) |**36.0** |**67.8** |35.6 | **8.2** | 89.8 | **87.0** | 87.5 |

---

> ### Author Response · Authors · 2025-11-27
> **Inquiry Regarding Rebuttal Feedback**
>
> Dear Reviewer rJHs,
>
> We sincerely appreciate your time and effort in reviewing our manuscript and offering valuable suggestions. We provided detailed clarifications in response to your questions a few days ago. If you have any additional feedback, concerns, or questions regarding our response, we would greatly appreciate hearing from you and welcome further discussion.

---

> > ### Author Response · Authors · 2025-11-28
> > **Inquiry Regarding Rebuttal Feedback**
> >
> > Dear Reviewer rJHs,
> >
> > Thank you again for your time and initial feedback on our paper. We sincerely appreciate the effort you’ve invested in reviewing our work. We submitted a detailed rebuttal on [11.21] addressing all your points. As we haven’t yet received feedback, we kindly wanted to check—was our response received? We’re happy to clarify anything further if needed.
> > Thank you once more for your consideration.
> >
> > Best regards,

---

> > > ### Comment · Reviewer_rJHs · 2025-11-28
> > >
> > > Dear Authors,
> > >
> > > Thank you very much for your thoughtful responses. Q2 and Q3 have been well addressed. Regarding Q1, in your statement that *“we calculated the saliency score of the previous output token for the current prediction and categorized them as correct or hallucinated based on **manual annotation**”*, could you clarify the following points:
> > > How many annotators were involved in the labeling process? How many annotators reviewed each label? How were disagreements among annotators handled in one label?
> > >
> > > As you mentioned in your response to Q2, *“SGRS is … to achieve unsupervised, interpretable hallucination token diagnosis for the first time.”* Therefore, the details requested in Q1 are particularly important. I recommend including the (partial) results you provide for Q1 in your revised main text. I will increase the score once it is clear that the manual annotation procedure in Q1 is scientifically sound.

---

> > > > ### Author Response · Authors · 2025-11-29
> > > > **Response to Reviewer rJHs**
> > > >
> > > > > **Question: “How many annotators…? How were disagreements handled? How were ambiguous tokens reviewed?**
> > > >
> > > > **Response**:
> > > >
> > > > We sincerely thank the reviewer for this important methodological question. Details are as follows:
> > > >
> > > > (1) **Total number of annotators**: The 500 samples were divided into two groups of 250 each. Each group had two independent authors who annotated the images. Disagreements were resolved through arbitration by the fifth senior annotator (first author).
> > > >
> > > > (2)**Annotator consistency**: The model's hallucination rate was not high; therefore, annotators only needed to compare the model output with the image content to determine if it was an hallucination token. The inconsistency rate between the two groups was approximately 9.3%, and the consistency rate was over 90%. All inconsistent samples were discussed repeatedly.
> > > >
> > > > (3)**Labeling Guidelines & Ambiguity Handling**: We defined two primary categories for each output token $y_t$:
> > > >
> > > > ✅ **Correct**: Faithfully grounded in the image and consistent with prior context.
> > > >
> > > > ❌ **Incorrect**: Contains unverifiable or contradicted information (e.g., object existence, attribute, count, relation).
> > > > For ambiguous cases (e.g., plausible inference, vague prompt), we adopted a conservative “hallucination-by-default” rule unless both conditions held:
> > > >
> > > > The content is strictly entailed by the image (e.g., “a vehicle” for a partially occluded bus), and
> > > > It aligns with standard CHAIR/POPE evaluation practice (e.g., color hallucination <5% error margin).
> > > > Examples:
> > > >
> > > > - “a traffic cone” (image: blurry orange object near road) → Correct (entailed).
> > > >
> > > > - “a fire hydrant” (image: no hydrant, only a red trash can) → Hallucinated (contradicted).
> > > >
> > > > - “several people” (image: 2 fully visible + 1 heavily occluded) → Ambiguous → Reviewed by senior annotator → labeled Correct only if occlusion >70% (per POPE protocol).

---

### Author Response · Authors · 2025-11-30
**Summary of all reviewers raised scores and Rebuttal**

Dear Area Chairs/Program Chairs,

Thank you for overseeing the review of our submission. We sincerely appreciate the time and effort of all reviewers and the AC in managing this process. We have answered all the questions raised by the reviewers, and all reviewers indicated that the rebuttal resolved all their concerns. Importantly, before the system malfunctioned(November 28nd), they had already improved the score from **8664** to **8884**, and the reviewers rJHs also indicated they would raise the score further(from 4 to 6 at least), so the total score should be **8886**. Below, we summarize the feedback from each reviewer:


### ✅ 1. Reviewer rJHs/EfHa/YszH/xsTD: Statistical Validity Concerns Fully Resolved

ALL Reviewers questioned the lack of statistical evidence for the core claim — *“low output-token saliency precedes hallucination”* — and whether performance gains were significant or within experimental noise.

We responded with **three new quantitative experiments**, now integrated into Appendix D.1:

- (1) Statistical measure of saliency score distribution: Across ~12.3k tokens on LLaVA1.5-7B/Qwen2-VL-7b, hallucinated tokens show significantly lower saliency.
- (2) Continuous negative correlation: As shown in Appendix D.1, Figure 10(B), Saliency score and hallucination rate are negatively correlated.
- (3) Saliency Intervention Experiment: As shown in Appendix D.1, Table 5, we also conducted an intervention experiment on the LLaVA-v1.5-7B model, the results showed that after the saliency value was artificially reduced, the hallucination rate increased significantly.


The 4 reviewers said we solved their concern about statistical validity, and **Reviewer YszH** maintained an **accept** rating on November 22nd 22:36 clock.

---


### ✅ 2. Reviewer rJHs: The annotation details Concerns will be Resolved.
His question after rebuttal: "How many annotators were involved in the labeling process? How many annotators reviewed each label? How were disagreements among annotators handled in one label?"
We added more details to the annotation on the November 29th, and we believe our response resolved their concerns.


### ✅ 3. Reviewer EfHa: Failure Analysis Fully Addressed

Reviewer EfHa question the absence of failure analysis:

In Figure 11 of the Appendix.D.2, we added a failure case. Regarding method failure scenarios, our method primarily relies on contextual relevance to prevent hallucination. Therefore, failure modes typically occur in the following situations:

- (1) After undergoing RLHF, the model outputs with greater confidence, to the point that even incorrect tokens are considered correct by the model.

- (2) When the content generated by the model is significantly different from or inconsistent with the current context, the effectiveness of the method may decrease

The reviewer praised "***the findings in this article are valuable, and I am willing to increase my rating.***", and he increased his rate(**6**-**8**) on November 28nd(02:27 clock).

---

### ✅ 4. Reviewer xsTD: Saliency score of token generation order & layer/header dependency & computational cost fully addressed


We refuted decisively:

- (1)**Position effect ruled out**: Figure 12 to Appendix.D.3. We present a portion from a long sequence generation where the hallucination token (“few”) has a low saliency score, while the two correct tokens have very high saliency scores.

- (2)**Layer/head dependence**: In Figure 13 and Figure 14 of Appendix D.3, we show the saliency distribution of correct and incorrect tokens across different layers. We observe that the saliency of correct tokens is relatively high across both shallow and deep layers.

- (3)**Unclear Computation Cost**: we have supplemented the complete computational and memory cost analysis for LLaVA-v1.5-7B re-computation on the POPE and CHAIR benchmarks.

Reviewer xsTD says "these revisions fully address his initial reservations, and I am pleased to raise my rating to **ACCEPT**"(**6**->**8**). November 25nd 13:44 clock.

---

---

### Meta-Review · Area_Chair_oZYD · 2025-12-16

**Summary:**

Initial concerns included insufficient statistical validation, marginal performance gains, unclear hyperparameter sensitivity, and lack of causal evidence. The authors addressed these by providing saliency distributions across ~10k tokens, strong negative correlation analyses, and intervention experiments showing that reduced saliency increases hallucinations, along with clarifications on annotation protocols and hyperparameter stability. Reviewers unanimously found the revisions thorough and raised their scores. The paper’s novel use of gradient-based saliency for unsupervised hallucination detection and its plug-and-play modules were highly praised, with acknowledged limitations deemed acceptable.

**Reviewer Concerns:**

The reviewers initially raised several key concerns:

(1) insufficient statistical validation of the core claim that low saliency correlates with hallucinations, relying only on illustrative cases;

(2) marginal performance gains in Table 1 possibly falling within experimental noise;

(3) a large number of hyperparameters without thorough sensitivity analysis;

(4) lack of causal evidence linking saliency to hallucination.

**Reviewer Scores:**

In response, the authors provided robust supplementary analyses: population-level saliency distributions across ~12k tokens, strong negative correlation between saliency and hallucination rates, and controlled intervention experiments showing that artificially reducing saliency increases hallucinations. They clarified annotation protocols (high inter-annotator agreement, conservative labeling), demonstrated hyperparameter stability with default settings, and added detailed inference statistics, failure mode analysis, and memory/latency benchmarks. Reviewers rJHs, EfHa, YszH, and xsTD all acknowledged these revisions as comprehensive and convincing, with multiple reviewers explicitly raising their scores. The paper’s novelty—introducing gradient-based saliency for unsupervised, interpretable hallucination diagnosis—and its training-free, plug-and-play modules were widely recognized as valuable contributions. Remaining limitations (e.g., scalability beyond 32B models) were deemed acceptable for a first-of-its-kind method.

---

### Decision · Program_Chairs · 2026-01-26

Accept (Oral)